# MEIOC prevents continued mitotic cycling and promotes meiotic entry during mouse oogenesis

Esther G. Ushuhuda[1,*], Jenniluyn T. Nguyen[2,*], Natalie G. Pfaltzgraff[1], Shelbie M. Wenner[1,3], Matthew Kofron[4,5] and Maria M. Mikedis[1,2,4,‡]

## ABSTRACT

To generate haploid gametes, germ cells must transition from mitosis to meiosis. In mammals, the transcriptional activator STRA8-MEIOSIN mediates the decision to enter the meiotic cell cycle, but how germ cells prevent continued mitotic cycling before meiotic entry remains unclear. MEIOC was previously shown to repress the mitotic program after meiotic entry. Here, we investigate the role of MEIOC in the mitosis-to-meiosis transition during mouse oogenesis. Using cell proliferation analysis and cell cycle transcriptomics, we demonstrate that MEIOC prevents continued mitotic cycling prior to meiotic entry in oogenic cells. We find that G1/S cyclin CCNA2 is downregulated during the mitosis-to-meiosis transition, and MEIOC contributes to this downregulation. MEIOC also promotes entry into meiosis by increasing *Meiosin* transcript abundance and consequently activating STRA8-MEIOSIN. Thus, in mouse oogenic cells, the transition from mitosis to meiosis occurs as two molecularly regulated steps – (1) the halt of mitotic cycling and (2) entry into the meiotic cell cycle – and that MEIOC modifies the cell cycle program to facilitate both steps in this transition.

KEY WORDS: MEIOC, STRA8, MEIOSIN, Meiosis, Fetal oocytes, Ovarian germ cell

## INTRODUCTION

Meiosis is the chromosomal foundation of sexual reproduction. During this specialized cell cycle, one round of DNA replication is followed by two rounds of chromosome segregation to produce haploid gametes. While the unique chromosomal events of meiosis (pairing, synapsis and crossing over of homologous chromosomes) are broadly conserved across eukaryotes, the mechanisms that drive the transition from mitosis to meiosis are less conserved (Kimble, 2011). The mitosis-to-meiosis transition requires a rewiring of the gene regulatory network to promote entry into the meiotic cell cycle

(Su et al., 2024). This rewiring is implemented at the G1/S transition, resulting in a meiotic S phase that is molecularly distinct from mitotic S phase (Pratto et al., 2021).

In mammals, the meiotic G1/S phase transition is induced in preleptotene oocytes and spermatocytes by retinoic acid signaling, which activates expression of the transcriptional regulator STRA8-MEIOSIN (Anderson et al., 2008; Baltus et al., 2006; Bowles et al., 2016, 2006; Ishiguro et al., 2020; Kojima et al., 2019; Koubova et al., 2014; Mark et al., 2008; Soh et al., 2015). During oogenesis, expression of *Meiosin* (i.e. *Gm4969*) is also activated by BMP signaling (Cheung et al., 2025). The STRA8-MEIOSIN heterodimer transcriptionally activates gene expression of G1/S cyclins *Ccne1*, *Ccne2* and *Ccna2*, DNA replication genes, and meiotic factors that coordinate the chromosomal events of meiotic prophase I. During oogenesis, STRA8 plays the additional function of sequestering RB1 to derepress E2F transcription factors and promote timely entry into meiotic S phase (Shimada et al., 2023). Based on studies of mice on the C57BL/6 genetic background, preleptotene oocytes and spermatocytes with genetically ablated *Stra8* or *Meiosin* fail to undergo meiotic DNA replication and progress into meiotic prophase I (Anderson et al., 2008; Baltus et al., 2006; Dokshin et al., 2013; Ishiguro et al., 2020). We note that the absence of completed DNA replication in *Stra8*- and *Meiosin*-null preleptotene oocytes (Anderson et al., 2008; Baltus et al., 2006; Dokshin et al., 2013; Ishiguro et al., 2020) also suggests that regulation independent of STRA8 and MEIOSIN prevents oogenic cells from continuing to cycle mitotically. However, a molecular basis for this regulation has yet to be defined.

We recently reported that, during spermatogenesis, the RNA-binding complex MEIOC-YTHDC2-RBM46 regulates STRA8-MEIOSIN by indirectly derepressing expression of *Meiosin* and other meiosis-associated genes (Pfaltzgraff et al., 2024). This activity enhances the molecular competence of mitotic spermatogenic cells to transition from mitosis to meiosis in response to retinoic acid. At the same time, this complex destabilizes *Ccna2* mRNA, resulting in the loss of CCNA2 protein expression and the establishment of a meiosis-specific cell cycle program in spermatocytes in meiotic prophase I (Pfaltzgraff et al., 2024; Soh et al., 2017). Both preleptotene oocytes and spermatocytes with *Meioc* and *Ythdc2* genetically ablated exhibit delayed progression into meiotic prophase I. They also prematurely attempt an abnormal metaphase, which coincides with their failure to properly establish a meiotic cell cycle program (Bailey et al., 2017; Soh et al., 2017). Genetic ablation of *Rbm46* also yields this phenotype during spermatogenesis, and while RBM46 is required for oogenesis, further work is needed to demonstrate that this requirement originates during the oogenic mitosis-to-meiosis transition (Peart et al., 2022). STRA8-MEIOSIN upregulates *Meioc* gene expression during meiotic entry (Ishiguro et al., 2020; Kojima et al., 2019; Soh et al., 2015). However, whether MEIOC and YTHDC2 act upstream of the STRA8-MEIOSIN

[1]Reproductive Sciences Center, Division of Developmental Biology, Cincinnati Children's Hospital Medical Center, Cincinnati, OH 45229, USA. [2]Whitehead Institute, Cambridge, MA 02142, USA. [3]Medical Scientist Training Program, University of Cincinnati College of Medicine, Cincinnati, OH 45267, USA. [4]Department of Pediatrics, University of Cincinnati College of Medicine, Cincinnati, OH 45267, USA. [5]Division of Developmental Biology, Cincinnati Children's Hospital Medical Center, Cincinnati, OH 45229, USA.
*These authors contributed equally to this work

‡Author for correspondence (maria.mikedis@cchmc.org)

S.M.W., 0000-0002-8951-9743; M.M.M., 0000-0001-7800-7120

complex to regulate the mitosis-to-meiosis transition during oogenesis remains unclear.

Here, we use analyses of cell proliferation and cell cycle transcriptomics to demonstrate that MEIOC halts mitotic cycling during oogenesis. We find that the oogenic meiotic cell cycle is molecularly distinguished from the mitotic cell cycle by the downregulation of G1/S cyclin CCNA2 at the transcript and protein levels. Consistent with its role in halting mitotic cycling, MEIOC downregulates CCNA2 protein expression. At the same time, MEIOC increases *Meiosin* transcript abundance and consequently activates the STRA8-MEIOSIN transcription factor to promote entry into meiotic S phase. BMP signaling similarly halts mitotic cycling and promotes meiotic entry through its upregulation of *Meioc* expression. Therefore, the transition from mitosis to meiosis occurs as two molecularly distinct steps – (2) the halt of mitotic cycling and (2) entry into the meiotic one – and MEIOC rewires the cell cycle program to facilitate both steps in this transition.

## RESULTS
### *Stra8* knockout oogenic cells arrest at the meiotic G1/S phase transition
STRA8 is required for entry into the meiotic cell cycle (i.e. meiotic G1-S transition), based on prior reports that used FACS analysis of DNA content to demonstrate that when wild-type oogenic cells are in meiotic S phase, *Stra8* knockout oogenic cells accumulate in G1

phase without undergoing DNA replication (Baltus et al., 2006). First, we sought to confirm these results using the incorporation of thymidine analog ethynyl deoxyuridine (EdU) as a marker of active DNA synthesis. We quantified fetal oogenic cells undergoing S phase within a 2 h window at embryonic day (E) 16.5, when wild-type oocytes are primarily in meiotic prophase I with a minority in meiotic S phase (Fig. 1A-C) (Soygur et al., 2021). As oogenic cells develop as a wave through the ovarian tissue, the E16.5 stage was chosen to ensure that all EdU-positive oogenic cells represent meiotic, and not mitotic, S phase. To confirm that EdU-positive oogenic cells at E16.5 reflected DNA synthesis rather than the repair of double-strand breaks during meiotic prophase I, we examined SYCP3-immunolabeled oogenic cells in zygotene and pachytene stages, when double-strand break repair occurs, and found that all of these cells were negative for EdU (Fig. S1, Table S1). For subsequent quantification of EdU-labeled DNA replication, oogenic cells were identified via immunostaining for DDX4 (also known as MVH), a germline-specific cytoplasmic RNA helicase that is expressed in oogenic cells from colonization of the embryonic gonads into adulthood (Toyooka et al., 2000).

We discovered that some *Stra8* knockout oogenic cells exhibited DNA replication, as they were marked by EdU (Fig. 1B). This phenotype is similar to that of *Stra8* knockout spermatocytes at the mitosis-to-meiosis transition (Anderson et al., 2008). We asked if *Stra8* knockout oogenic cells exhibited EdU labeling at a similar

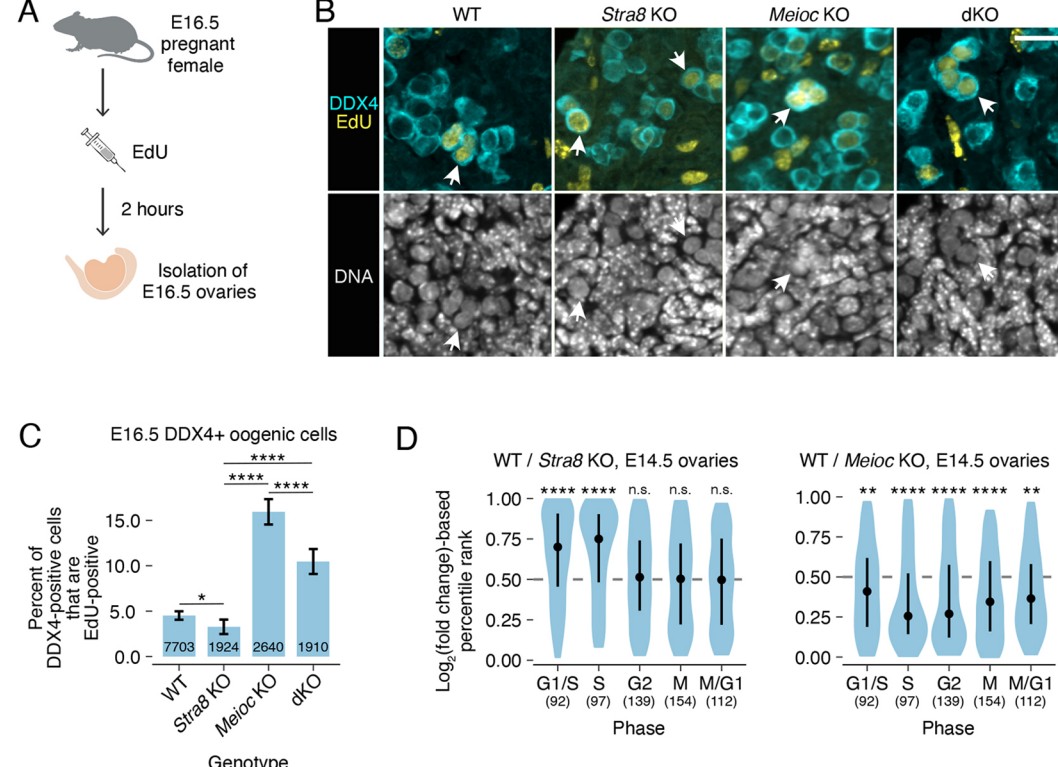

**Fig. 1. *Meioc* knockout oogenic cells continue to mitotically cycle when wild-type oogenic cells have entered meiosis.** (A) Schematic of EdU labeling experiment in E16.5 ovaries. (B) Fluorescent labeling of exemplary EdU in DDX4-positive oogenic cells (arrows) from wild-type, *Stra8* knockout, *Meioc* knockout and *Meioc Stra8* double knockout (dKO) ovaries at E16.5. *Meioc Stra8* double knockout panel also appears in Fig. 3C. Scale bar: 20 μm. (C) Percentage of DDX4-positive oogenic cells that are positive for EdU in E16.5 ovaries. Error bars represent 95% confidence interval. Numbers on bars represent the total number of DDX4-positive cells quantified. Data were collected from five wild-type embryos, five *Stra8* knockout embryos, three *Meioc* knockout embryos and three *Meioc Stra8* double knockout embryos. (D) Log$_2$-fold change-based percentile ranks of genes associated with cell cycle phases for wild type versus *Stra8* knockout and wild type versus *Meioc* knockout from whole ovary RNA-seq data. Dot represents the median; whiskers represent the interquartile range. The numbers of genes per cell cycle phase are in parentheses. *adj. $P<0.05$; **adj. $P<0.01$; ****adj. $P<0.0001$; n.s., not significant. See Table S1 for statistical details.

frequency to wild-type oogenic cells, and found that *Stra8* knockout oogenic cells were less likely to be EdU positive (Fig. 1C, Table S1). This DNA replication in *Stra8* knockout oogenic cells does not represent meiotic S phase because meiotic cohesin REC8 is not loaded onto the chromosomes (Baltus et al., 2006). While Baltus et al. (2006) found that *Stra8* knockout oogenic cells accumulated in G1 phase and not in S phase or G2/M, their results relied on FACS analysis of DNA content, which is less sensitive than EdU labeling for identifying cells in S phase. Therefore, our EdU-based results demonstrate that some *Stra8* knockout oogenic cells can initiate DNA replication, while the DNA content analysis of Baltus et al. (2006) demonstrates that these cells do not complete DNA replication. The molecular activity of STRA8 is consistent with regulation of the meiotic G1/S phase transition, as STRA8 directly activates expression of G1/S regulators and DNA replication genes, as well as interacts with RB1 to sequester it from E2F transcription factors (Ishiguro et al., 2020; Kojima et al., 2019; Shimada et al., 2023). Therefore, we conclude that, in the absence of *Stra8*, the majority of oogenic cells arrest at the meiotic G1/S phase transition but some cells slip into early S phase.

To further confirm that *Stra8* knockout oogenic cells fail to complete meiotic S phase, we re-analyzed bulk RNA-seq data of wild-type and *Stra8* knockout E14.5 ovaries (Soh et al., 2017) and examined the abundance of transcripts associated with specific phases of the cell cycle. At E14.5, the wild-type ovary contains mitotic oogonia, preleptotene oocytes (in G1 and meiotic S phase preceding meiotic prophase I), and oocytes in early meiotic prophase I. We found that STRA8 upregulates the abundance of cell cycle transcripts associated with G1/S and S phase, but not G2, M or M/G1 (Fig. 1D). This analysis confirms that genetic ablation of *Stra8* causes oogenic cells to arrest primarily at the meiotic G1/S phase transition.

### MEIOC prevents continued mitotic cycling in preleptotene oocytes

Observations that *Stra8* knockout oogenic cells arrest at the meiotic G1/S transition demonstrate that these cells are no longer mitotically cycling. Therefore, regulation independent of STRA8 prevents these cells from proceeding with a mitotic cell cycle. MEIOC represses the mitotic cell cycle program after oocytes have entered meiotic prophase I (Soh et al., 2017). Here, we hypothesized that MEIOC also acts prior to meiotic entry to prevent continued mitotic cycling during oogenesis.

First, we tested our hypothesis at the level of transcriptomics. If MEIOC prevents continued mitotic cycling during oogenesis, then MEIOC would downregulate cell cycle-associated gene expression across all phases of the cell cycle. Alternatively, if MEIOC is required for efficient progression through meiotic S phase, as previously concluded, without affecting mitotic cycling, then we would predict that MEIOC upregulates cell cycle-associated gene expression at G1/S and S phases, like STRA8. Re-analysis of wild-type versus *Meioc* knockout bulk RNA-seq data from E14.5 ovaries (Soh et al., 2017) revealed that MEIOC downregulates the abundance of transcripts across all cell cycle phases (Fig. 1D), a pattern that was distinct from that observed in the STRA8 analysis. This indicates that when wild-type oogenic cells undergo the meiotic G1/S phase transition, *Meioc*-knockout oogenic cells continue to cycle mitotically.

Next, we tested our hypothesis via cell cycle analysis. We reasoned that if MEIOC halts mitotic cycling in oogenic cells, then loss of *Meioc* would allow *Stra8*-null oogenic cells to continue proliferating mitotically, resulting in an increase in EdU-positive oogenic cells

relative to *Stra8* single knockout. Alternatively, if MEIOC does not halt mitotic cycling prior to meiotic entry, then *Meioc Stra8* double-knockout and *Stra8* single-knockout ovaries would exhibit a similar number of EdU-positive oogenic cells halted at the meiotic G1/S phase transition. Here, we found that *Meioc Stra8* double-knockout oogenic cells exhibited increased synthesized DNA relative to *Stra8* knockout cells (Fig. 1B,C; Table S1). Therefore, MEIOC prevents continued mitotic cycling in preleptotene oocytes.

### The mitotic cell cycle program is repressed in preleptotene oocytes in G1 phase

The model that MEIOC blocks continued mitotic cycling implies that the cell cycle program in G1 phase in preleptotene oocyte is molecularly distinct from that in mitotic oogonia. To identify these differences at the transcriptomic level, we re-analyzed scRNA-seq from E14.5 oogenic cells, which were isolated via sorting for the *Pou5f1*:EGFP reporter that identifies germ cells in embryonic gonads (Szabó et al., 2002; Zhao et al., 2020). Using cell type-enriched marker expression and transcriptome-based cell cycle analysis, the following oogenic cell types were identified: one cluster in mitosis [mitotic oogonia (Mit)]; three clusters spanning the transition from mitosis to meiosis [preleptotene oocytes in G1, early S and late S phase (pL G1, pL eS and pL lS, respectively]; and two clusters in meiotic prophase [leptotene (L) and zygotene (Z) oocytes; Fig. 2A, Fig. S2A-F; Tables S2, S3). The relative proportions of these oogenic stages replicated prior histological analysis of the E14.5 ovary (Fig. S2C; Arora et al., 2016). Cell cycle designations were further confirmed by the downregulation of *Mki67* in the pL G1 cluster and upregulation of replication-dependent histones in the pL eS and lS clusters (Fig. S2E,F).

We sought to define how the pL G1 cluster differs from the Mit and pL eS clusters. We carried out differential expression analysis between sequential clusters, and identified genes whose transcript abundance is increased ($\log_2$ fold change pL G1/Mit>0 and adjusted $P<0.05$) or decreased ($\log_2$ fold change pL G1/Mit<0 and adjusted $P<0.05$) in the later cluster of each sequential pair. Based on functional analysis, GO annotation 'meiotic cell cycle' was enriched among genes upregulated during progression from Mit to pL G1 clusters, and even further enriched from pL G1 to pL eS clusters (Fig. 2B). GO annotation 'mitotic cell cycle process' was enriched among genes downregulated during progression from Mit to pL G1 clusters, but not from pL G1 to pL eS clusters. This indicates that inhibition of the mitotic cell cycle and meiosis-associated differentiation occur in preleptotene cells in G1 phase, before entry into meiotic S phase.

Given that *Meioc* and *Stra8* regulate the cell cycle in preleptotene oocytes, we examined their transcript abundance across the E14.5 oogenic trajectory. Oogenic cells progressively upregulated both *Meioc* and *Stra8* as they transitioned from the Mit to pL G1 to pL eS to pL lS clusters (Fig. 2C). *Meioc* also remained highly abundant through early meiotic prophase I, while *Stra8* was downregulated during the transition from L to Z clusters. In contrast, *Rbm46* and *Ythdc2*, which are binding partners of *Meioc*, exhibited more consistent expression levels across this trajectory. [While *Meiosin*, the binding partner of *Stra8*, is expressed in preleptotene oocytes (Ishiguro et al., 2020; Shimada et al., 2023), it was not detectable in this dataset, as the technical limitations of scRNA-seq oftentimes prevents detection of lowly expressed genes.] Therefore, changes in mitotic and meiotic cell cycle transcripts in preleptotene oocytes is associated with upregulated expression of *Meioc* and *Stra8*.

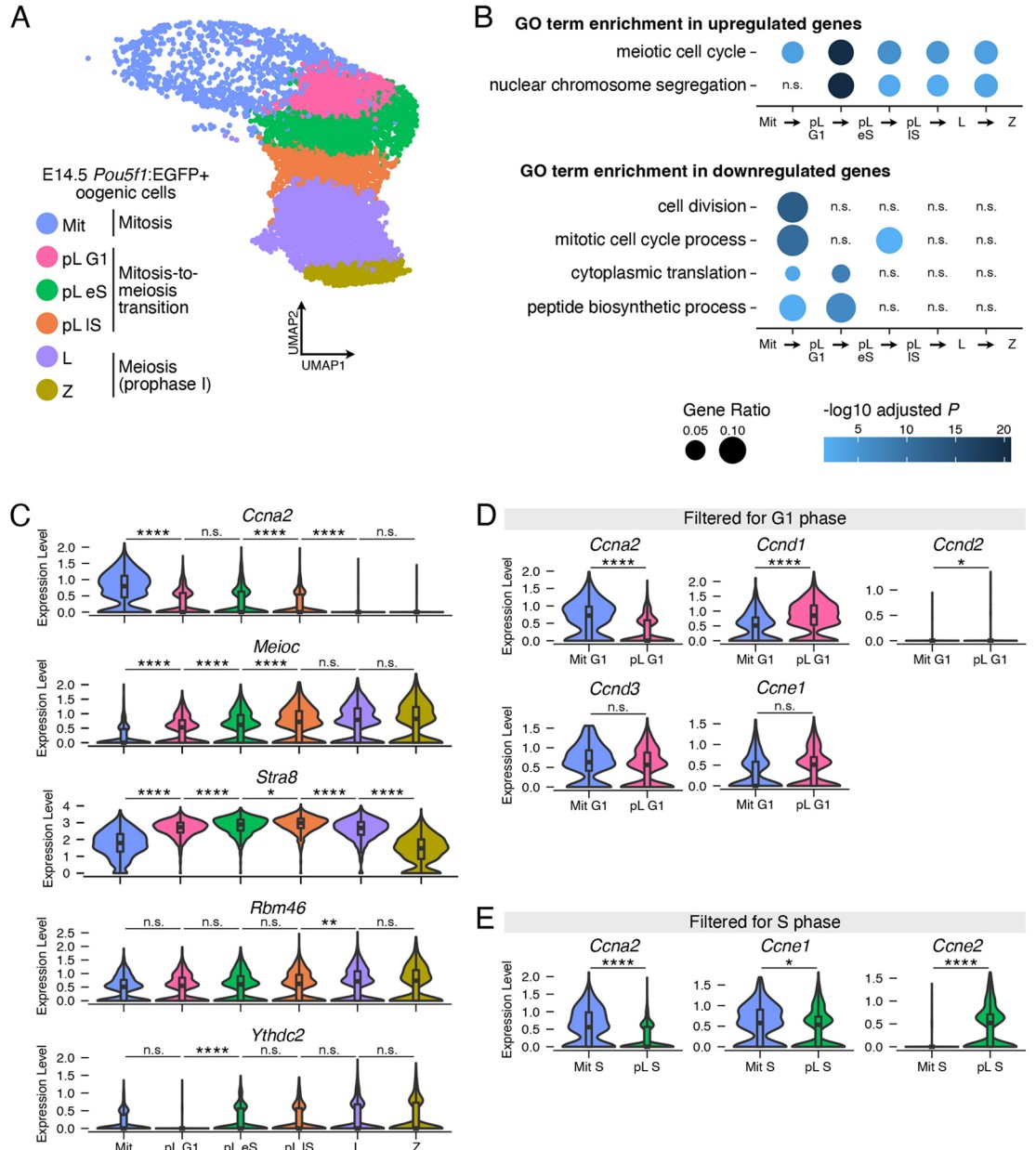

**Fig. 2. Preleptotene oocytes repress the mitotic cell cycle program and downregulate G1/S cyclin *Ccna2* in G1 phase before meiotic S phase.**
(A) UMAP visualization of E14.5 wild-type oogenic cell clusters in re-analysis of scRNA-seq data. (B) Gene Ontology (GO) term enrichment analysis for sequential pairwise comparisons of oogenic cell clusters. Upregulated and downregulated genes are those with increased and decreased transcript abundance, respectively, as cells move from the earlier to later cluster. GO terms represent the top category for upregulated genes in the pL G1 versus Mit comparison; the top category for upregulated genes in the pL eS versus pL G1 comparison; the top two categories for downregulated genes in the pL G1 versus Mit comparison; and the top two categories for downregulated genes in the pL eS versus pL G1 comparison. (C) Expression levels of selected genes in oogenic cell clusters. (D) Expression levels of cyclins that regulate progression through G1 phase in oogenic cells at G1 phase. Mit G1, Mit cells filtered for G1 phase. In these graphs only, pL G1 cells were similarly filtered for G1 phase. (E) Expression levels of cyclins that regulate progression through G1 phase in oogenic cells at S phase. Mit S, Mit cells filtered for S phase; pL S, pL eS and IS cells filtered for S phase. *adj. *P*<0.05; **adj. *P*<0.01; ****adj. *P*<0.0001; n.s., not significant. See Table S2 for statistical details.

## Repression of the mitotic cell cycle program is associated with downregulation of CCNA2

We sought to identify a molecular basis for how the mitotic cell cycle is inhibited in preleptotene oocytes in G1 phase. In budding yeast, entry into meiotic S phase is preceded by the repression of cyclins that regulate G1 phase (Colomina et al., 1999; Su et al., 2024). As these cyclins are, in part, regulated transcriptionally, we hypothesized that preleptotene oocytes in G1 phase downregulate cyclin transcript abundance. To test this, we identified mitotic and preleptotene cells in G1 phase by filtering for cells designated as 'G1 phase' in the Mit and pL G1 clusters. We then identified differentially abundant transcripts between the two groups. We focused this analysis on cyclins that regulate progression through G1 phase: cyclin D, cyclin E and cyclin A (D'Urso et al., 1990; Girard et al., 1991; Pagano et al., 1992; Resnitzky et al., 1994; Rosenblatt et al., 1992; Sprenger et al., 1997). Of these, *Ccna2* was downregulated in preleptotene oocytes versus mitotic oogonia in G1 phase (Fig. 2D). By contrast, *Ccnd1* and *Ccnd2* were upregulated.

Additional cyclins *Ccne1* and *Ccnd3* exhibited unchanged abundances. Therefore, repression of the mitotic cell cycle in preleptotene oocytes in G1 phase involves the downregulation of transcript abundance of *Ccna2*, which regulates the G1/S phase transition of the cell cycle (Donjerkovic and Scott, 2000).

CCNA2 also regulates progression through S phase. We therefore asked if preleptotene oocytes in meiotic S phase exhibit reduced *Ccna2* abundance relative to oogonia in mitotic S phase. To test this, we identified differentially abundant transcripts between mitotic oogonia and preleptotene oocytes in S phase (cells in Mit versus pL eS/lS clusters annotated as 'S phase'). *Ccna2* was downregulated in preleptotene oocytes versus mitotic oogonia in S phase (Fig. 2E). The additional S phase cyclins *Ccne1* and *Ccne2* are downregulated and upregulated, respectively.

During spermatogenesis, *Ccna2* is downregulated during the transition from mitosis to meiosis, such that mitotic spermatogonia express high levels of *Ccna2* while spermatocytes in meiotic prophase I express low levels (Ravnik and Wolgemuth, 1996). Here, we asked if *Ccna2* is similarly regulated during oogenesis. Differential abundance analysis of sequential germ cell clusters demonstrated that *Ccna2* is progressively downregulated into meiotic prophase I (Fig. 2C). Therefore, oogenic cells transitioning from mitosis to meiosis downregulate *Ccna2* transcript abundance, similar to spermatogenic cells.

Next, we hypothesized that the downregulation of *Ccna2* transcript abundance during the transition from mitosis to meiosis leads to reduced CCNA2 protein expression. To test this, we immunostained ovarian tissue sections and quantified fluorescent intensity of CCNA2. We examined ovarian germ cells at E12.5, representing mitotic cells; at E14.5, when germ cells range from mitosis to early meiotic prophase I; and at E16.5, when germ cells are in late meiotic S phase through prophase I. As CCNA2 levels fluctuate during the cell cycle, we quantified EdU-positive and -negative oogenic cells in S phase and outside S phase separately. We found that, for oogenic cells in and outside S phase, mitotic oogonia at E12.5 exhibited the highest levels of CCNA2 protein, while meiotic oocytes at E16.5 exhibited the lowest (Fig. 3A,B; Table S4). Therefore, CCNA2 protein is downregulated as mitotic oogonia transition to meiotic oocytes.

## MEIOC and STRA8 downregulate CCNA2 protein expression during the transition from mitosis to meiosis

During spermatogenesis, MEIOC destabilizes *Ccna2* mRNA and consequently downregulates CCNA2 protein expression by meiotic prophase I (Pfaltzgraff et al., 2024; Soh et al., 2017). Here, we asked whether MEIOC also downregulates CCNA2 during oogenesis. We compared oogenic cells in wild-type and *Meioc* knockout ovaries, and found that *Meioc* knockout cells exhibited higher levels of CCNA2 signal intensity for both EdU-positive and -negative populations (Fig. 3C,D; Table S4). The difference between the wild-type and *Meioc* knockout cells was small, which is consistent with a role for MEIOC in destabilizing *Ccna2* mRNA during spermatogenesis (Pfaltzgraff et al., 2024). At the same time, other molecular changes may also contribute to this change in protein levels. Therefore, MEIOC downregulates CCNA2 protein expression during oogenesis.

Next, we asked if STRA8 affects CCNA2 expression. Given that mitotic oogonia exhibit higher levels of CCNA2 than meiotic oocytes, we would predict that E16.5 *Stra8* knockout preleptotene oocytes halted in meiotic G1/S would exhibit higher levels of CCNA2 than wild-type oocytes in late meiotic S phase or meiotic prophase I. Our comparison of CCNA2 intensity revealed that *Stra8* knockout oogenic cells exhibited higher levels of CCNA2 than

wild-type oocytes (Fig. 3C,D). This was true for EdU-positive and -negative oocytes. Therefore, STRA8 is required to downregulate CCNA2 protein expression, likely as part of the overall transition from mitosis to meiosis.

Finally, we asked if MEIOC affects CCNA2 protein levels in the *Stra8* knockout background. By comparing EdU-positive oogenic cells in *Stra8* knockout and dKO ovaries, we found that dKO cells exhibited higher CCNA2 levels than *Stra8* knockout cells (Fig. 3C,D). This demonstrates that MEIOC downregulates CCNA2 levels in a *Stra8* knockout background, as observed on a wild-type *Stra8* background. However, a comparison of EdU-negative oogenic cells revealed lower levels of CCNA2 in the dKO than in the *Stra8* knockout ovary. As the *Stra8* knockout oogenic cells are arrested at the G1/S phase transition, while dKO oogenic cells continue to cycle mitotically, we propose that this difference in CCNA2 levels reflects discrepancies in cell cycle phase rather that the direct effects of MEIOC activity. In total, we conclude that, in *Stra8* knockout oogenic cells in S phase, MEIOC represses CCNA2 to prevent continued mitotic cycling.

## MEIOC links halted mitotic cycling to meiotic entry by upregulating *Meiosin*

With our newfound understanding that MEIOC prevents continued mitotic cycling while STRA8-MEIOSIN promotes entry into the meiotic cell cycle, we considered how halted mitotic cycling may be molecularly linked to meiotic entry. We recently reported that, during spermatogenesis, MEIOC is required to upregulate *Meiosin* gene expression (Pfaltzgraff et al., 2024). Here, we hypothesized that MEIOC also upregulates *Meiosin* during oogenesis. In re-analysis of bulk RNA-seq data from wild-type and *Meioc*-null ovaries, we confirmed that MEIOC upregulates *Meiosin* (Fig. 4A; Table S5). As MEIOSIN is required for meiotic S phase, the regulation of *Meiosin* expression by MEIOC impacts meiotic entry during oogenesis.

STRA8-MEIOSIN acts as an obligate heterodimer that transcriptionally activates gene expression at meiotic entry. As previously reported, *Stra8* was downregulated by MEIOC in the fetal ovary (Fig. 4A; Soh et al., 2017). We hypothesized that the upregulation of *Meiosin* gene expression by MEIOC activates STRA8-MEIOSIN transcriptional changes, even with reduced *Stra8* expression. To test this, we asked whether genes upregulated and downregulated by MEIOC were also dependent on STRA8, using bulk RNA-seq data from wild-type and *Stra8* knockout E14.5 ovaries. We found MEIOC-upregulated genes ($\log_2$ fold change>0 and adjusted $P<0.05$) exhibited a statistically significant overlap with STRA8-upregulated genes ($\log_2$ fold change>0 and adjusted $P<0.05$; Fig. 4B). Notably, this overlap included *Meiosin*. We also identified a small overlap between the MEIOC-downregulated and STRA8-downregulated genes sets ($\log_2$ fold change<0 and adjusted $P<0.05$) that was statistically significant. We conclude that a significant fraction of the MEIOC-dependent changes in transcript abundance are due to activation of the STRA8-MEIOSIN transcriptional program.

Next, we asked if MEIOC preferentially impacts genes that are directly targeted by STRA8-MEIOSIN in oogenic cells. Based on ChIP-seq data from preleptotene spermatocytes, STRA8-MEIOSIN exhibits sequence-specific binding to gene promoters (Ishiguro et al., 2020; Kojima et al., 2019). We reasoned that genes (1) upregulated by STRA8 in fetal ovary RNA-seq dataset (Fig. 4B) and (2) whose promoters are bound by STRA8, as defined by ChIP-seq in preleptotene spermatocytes (Kojima et al., 2019), were likely activated by STRA8 in oogenic cells. Among STRA8-upregulated

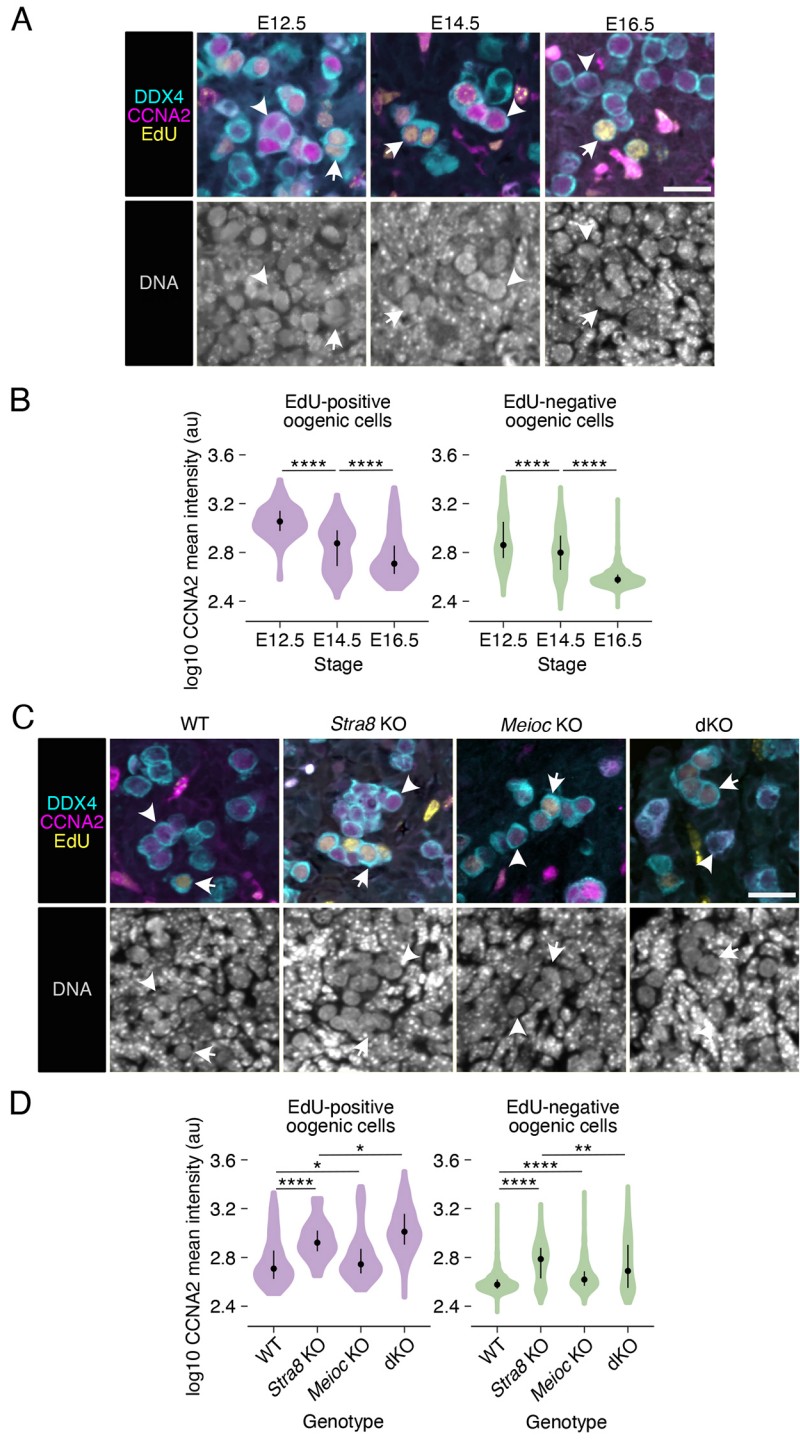

**Fig. 3. CCNA2 protein expression is downregulated during the transition from mitosis to meiosis by MEIOC and STRA8.** (A) Fluorescent labeling of CCNA2 in DDX4-positive oogenic cells in mitotic stages (E12.5), transitioning from mitosis to meiosis (E14.5) and in meiotic stages (E16.5). Arrows and arrowheads mark EdU-positive and -negative oogenic cells, respectively. Scale bar: 20 µm. (B) Quantification of CCNA2 intensity in EdU-positive and -negative oogenic cells. Data for EdU-positive cells represents 161 cells from three embryos at E12.5, 240 cells from five embryos at E14.5 and 190 cells from five embryos at E16.5. Data for EdU-negative cells represents 448 cells from three embryos at E12.5, 925 cells from five embryos at E14.5 and 2492 cells from five embryos at E16.5. Dot represents the median; whiskers represent the interquartile range. (C) Labeling of CCNA2 in DDX4-positive oogenic cells from wild-type, *Stra8* knockout, *Meioc* knockout and *Meioc Stra8* double knockout (dKO) ovaries at E16.5. Arrows and arrowheads mark EdU-positive and -negative oogenic cells, respectively. *Meioc Stra8* double knockout panel also appears in Fig. 1B. Scale bar: 20 µm. (D) Quantification of CCNA2 intensity in EdU-positive and -negative oogenic cells. Data for EdU-positive cells represents 190 cells from five wild-type embryos, 35 cells from five *Stra8* knockout embryos, 144 cells from three *Meioc* knockout embryos and 124 cells from three *Meioc Stra8* double knockout embryos. Data for EdU-negative cells represents 2492 cells from five wild-type embryos, 579 cells from five *Stra8* knockout embryos, 909 cells from three *Meioc* knockout embryos and 296 cells from three *Meioc Stra8* double knockout embryos. Dot represents the median; whiskers represent the interquartile range. *adj. $P<0.05$; **adj. $P<0.01$; ****adj. $P<0.0001$. See Table S4 for statistical details.

genes in oogenic cells, 75% were bound by STRA8 at their promoters in testis data and therefore are likely directly activated by STRA8 in oogenic cells (Fig. 4C). Among MEIOC-upregulated genes, 20% were activated by STRA8. By contrast, STRA8-activated genes represented only 1% of expressed genes. This analysis supports the model that the activity of MEIOC leads to increased STRA8-MEIOSIN activity.

Our hypothesis that the activity of MEIOC leads to STRA8-MEIOSIN-mediated transcriptional changes also predicts that the changes in transcript abundance of STRA8-dependent genes would be correlated to MEIOC-dependent transcript abundance changes.

By comparing the $\log_2$-fold changes due to MEIOC and STRA8, we found that the STRA8-dependent genes (adjusted $P<0.05$) exhibited significantly correlated effects in the MEIOC and STRA8 datasets (Fig. 4D). Given that the *Meioc* knockout exhibits prolonged mitotic cycling and reduced *Meiosin* levels (Figs 1C,D and 4A), this analysis supports the model that the loss of MEIOC leads to reduced STRA8-MEIOSIN activity in E14.5 oogenic cells.

In the absence of MEIOC, *Meiosin* is still expressed, but at lower levels than in wild-type controls. In this scenario, we would predict that transcriptomic changes in the MEIOC dataset would be less extreme than those in the STRA8 dataset. To test this, we focused on

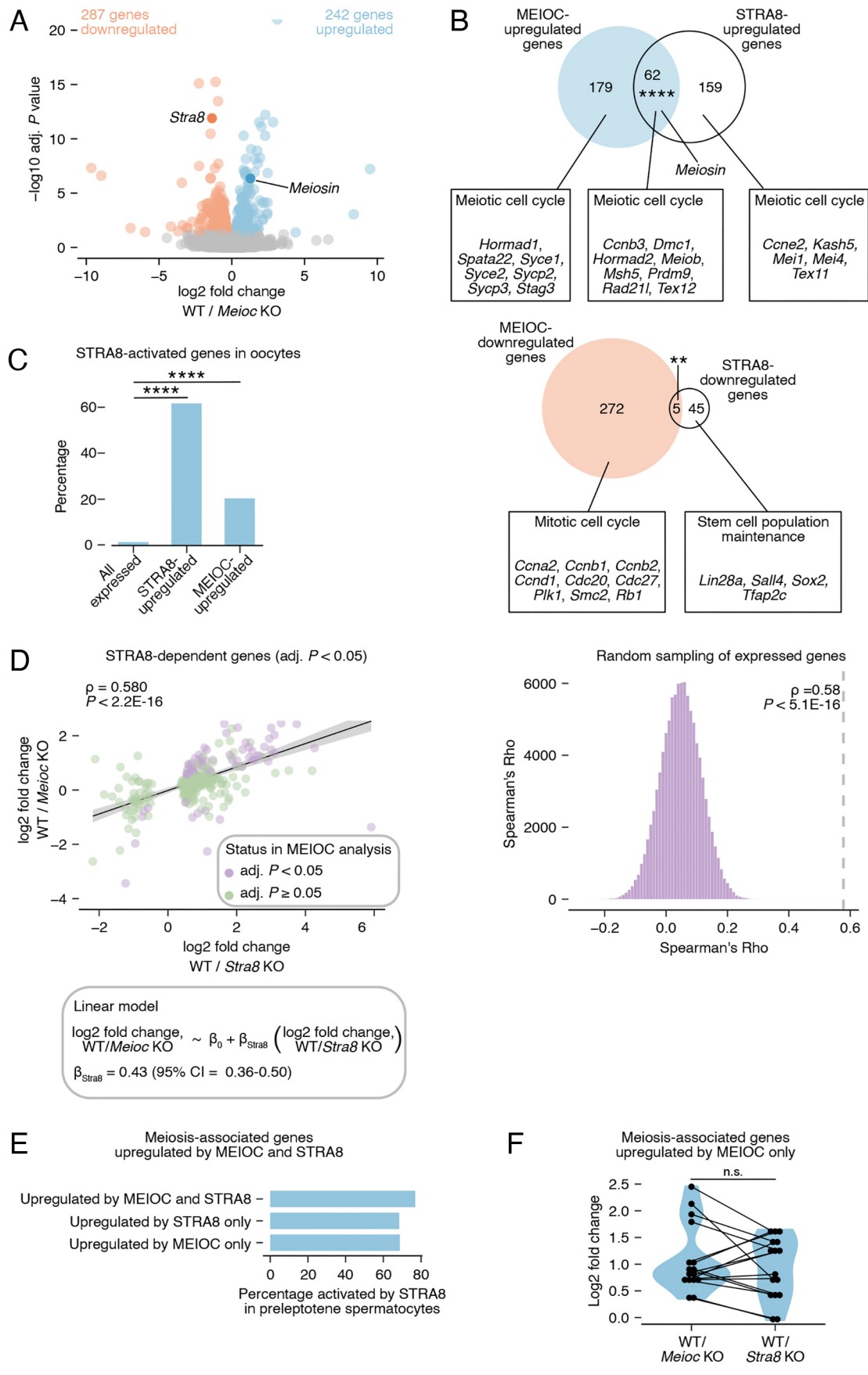

**Fig. 4.** See next page for legend.

**Fig. 4. MEIOC upregulates *Meiosin* expression and activates the STRA8-MEIOSIN transcription factor to support meiotic entry.**
(A) Volcano plot of differential expression results from re-analysis of RNA-seq data generated from E14.5 wild-type and *Meioc* knockout whole ovaries. (B) Venn diagrams comparing gene sets. STRA8-upregulated genes are derived from re-analysis of RNA-seq data generated from E14.5 wild-type and *Stra8* knockout whole ovaries. Boxes highlight select Gene Ontology terms and associated genes enriched within each gene set. *Meiosin* is not yet annotated as a 'meiotic cell cycle' gene and is highlighted separately. (C) Percentage of STRA8-activated genes among all expressed genes, STRA8-upregulated genes and MEIOC-upregulated genes in E14.5 ovaries. STRA8-activated genes were defined as those that are upregulated by STRA8 in E14.5 ovaries and bound at the promoter by STRA8 in preleptotene spermatocytes, as defined by Kojima et al. (2019). (D) Left: correlation between MEIOC and STRA8 RNA-seq analyses of whole ovaries at E14.5. Analysis is restricted to STRA8-upregulated and -downregulated genes (adjusted *P*<0.05). *P*-value represents the probability that Spearman rho does not equal 0. $\beta_{Stra8}$ represents slope in the linear model. Right: distribution of correlations for gene sets obtained by random sampling of genes expressed in MEIOC and STRA8 RNA-seq analyses. *P*-value represents that probability of obtaining an equal or larger correlation by random sampling. (E) Percentage of meiosis-associated genes upregulated by MEIOC and STRA8 in E14.5 ovaries that are also activated by STRA8 in preleptotene spermatocytes, as reported by Kojima et al. (2019). (F) Log$_2$-fold change induced by MEIOC and STRA8 in meiosis-associated genes that are statistically upregulated by MEIOC only. **adj. *P*<0.01; ****adj. *P*<0.0001. See Table S5 for statistical details.

STRA8-dependent genes and modeled their MEIOC-dependent log$_2$-fold changes in transcript abundance as a linear function of their STRA8-dependent log$_2$ fold changes. In this model, the slope of regression ($\beta_{Stra8}$) represents the MEIOC-dependent log$_2$ fold change in transcript abundance for every STRA8-dependent log$_2$ fold change of 1. If loss of MEIOC causes complete loss of STRA8-MEIOSIN activity, then the slope would be equal to 1. Alternatively, if loss of MEIOC reduces but does not completely ablate STRA8-MEIOSIN activity, then the slope would fall between 0 and 1. The linear model produced a slope ($\beta_{Stra8}$) of 0.43 (95% confidence interval: 0.36-0.50; Fig. 4D), which indicates that loss of MEIOC results in reduced STRA8-MEIOSIN activity. Therefore, MEIOC is required to upregulate STRA8-MEIOSIN activity, but some STRA8-MEIOSIN activity persists in the absence of MEIOC. This persistent STRA8-MEIOSIN activity is consistent with the *Meioc* knockout phenotype, in which oogenic cells enter meiotic prophase I on a delayed timeline (Abby et al., 2016; Soh et al., 2017).

Next, we considered the biological processes enriched in gene sets regulated by MEIOC and STRA8. Genes upregulated by both MEIOC and STRA8, MEIOC only, and STRA8 only were all enriched for Gene Ontology terms related to the meiotic cell cycle (Fig. 4B). While 16 meiosis-associated genes were identified as upregulated by MEIOC only, we noted that many of these genes were activated by STRA8 (i.e. bound by STRA8 at their promoters and upregulated by STRA8) during spermatogenesis (Fig. 4E; Kojima et al., 2019). In addition, MEIOC- and STRA8-dependent fold changes for these 16 genes as a group were statistically similar (Fig. 4F). This indicates that most of the meiosis-associated genes upregulated by MEIOC only are likely upregulated directly by STRA8, but they have failed to reach statistical significance in the STRA8 bulk RNA-seq dataset. As MEIOC does not target meiosis-associated mRNAs (Soh et al., 2017), we conclude that MEIOC supports the upregulation of meiosis-associated genes through the activity of STRA8.

Genes downregulated by MEIOC only included *Ccna2* and were enriched for Gene Ontology terms associated with the mitotic cell cycle, consistent with the role of MEIOC in halting mitotic cycling

(Fig. 4B). By contrast, genes downregulated by STRA8 only were enriched for genes associated with stem cell maintenance. As STRA8 does not directly bind the promoters of these genes in preleptotene spermatocytes (Kojima et al., 2019), their downregulation by STRA8 in fetal ovaries likely occurs indirectly upon entry into the meiotic cell cycle. This analysis supports our EdU-based proliferation results demonstrating that MEIOC halts mitotic cycling independently of STRA8. In total, these analyses demonstrate that MEIOC links the halt in mitotic cycling to entry into the meiotic cell cycle by promoting the upregulation of *Meiosin* transcript abundance and STRA8-MEIOSIN activity.

## BMP signaling upregulates MEIOC to halt mitotic cycling

A recent study demonstrated that loss of BMP signaling, via conditional deletion of the BMP receptor *Bmpr1a*, caused oogenic cells to continue mitotic cycling and delay meiotic entry, based on increased expression G2/M marker phospho-histone H3 and S phase label BrdU in *Bmpr1a* conditional knockout ovarian sections (Cheung et al., 2025). Here, we sought to confirm these results via cell cycle transcriptomics. We re-analyzed RNA-seq data from wild-type and *Bmpr1a* conditional knockout ovaries at E14.5, and examined the abundance of cell cycle transcripts. We found that BMPR1A downregulates the abundance of transcripts associated with all phases of the cell cycle, like MEIOC (Fig. 5A; Table S6). This analysis confirms that BMP signaling directs oogenic cells to halt their mitotic cycling prior to meiotic entry.

As previously reported, *Meioc* gene expression was upregulated by BMP signaling (Fig. 5B; Cheung et al., 2025). Given that MEIOC halts mitotic cycling in oogenic cells, we considered whether BMP signaling controls mitotic cycling through MEIOC. If true, then we would predict that MEIOC-dependent changes in the transcriptome would be evident in the changes induced by BMP signaling. We tested this via two approaches. First, we asked whether MEIOC-dependent genes were also dependent on BMP signaling. We found a statistically significant overlap between MEIOC-upregulated and BMPR1A-upregulated gene sets, as well as between MEIOC-downregulated and BMPR1A-downregulated gene sets (Fig. 5C). Second, we asked whether transcript abundance changes induced by BMP signaling were correlated to those induced by MEIOC. By comparing the log$_2$-fold changes due to BMPR1A and MEIOC, we found that the MEIOC-dependent genes (adjusted *P*<0.05) exhibited significantly correlated effects in the BMPR1A and MEIOC datasets (Fig. 5D). We conclude that a significant fraction of the transcript abundance changes induced by BMP signaling are due to the activity of MEIOC.

Loss of BMP signaling in oogenic cells yields reduced, but not completely ablated, *Meioc* expression, despite near complete loss of BMPR1A protein expression in histological sections (Table S6; Cheung et al., 2025). If MEIOC was inducing transcriptomic changes in the BMPR1A-dependent program, then we would predict that conditional deletion of *Bmpr1a* would have a smaller effect on MEIOC-dependent genes than MEIOC. Alternatively, if the BMPR1A-mediated change in MEIOC-dependent genes was larger than that induced by MEIOC, then BMP signaling is likely regulating these genes independently of MEIOC. We tested this by modeling the BMPR1A-dependent log$_2$-fold changes in transcript abundance of MEIOC-dependent genes as a linear function of their MEIOC-dependent log$_2$-fold changes. In this model, the slope of regression ($\beta_{Meioc}$) represents the BMPR1A-dependent log$_2$-fold change in transcript abundance for every MEIOC-dependent log$_2$-fold change of 1. Given that the *Bmpr1a* conditional knockout model induces partial loss of *Meioc* expression, we predict the slope

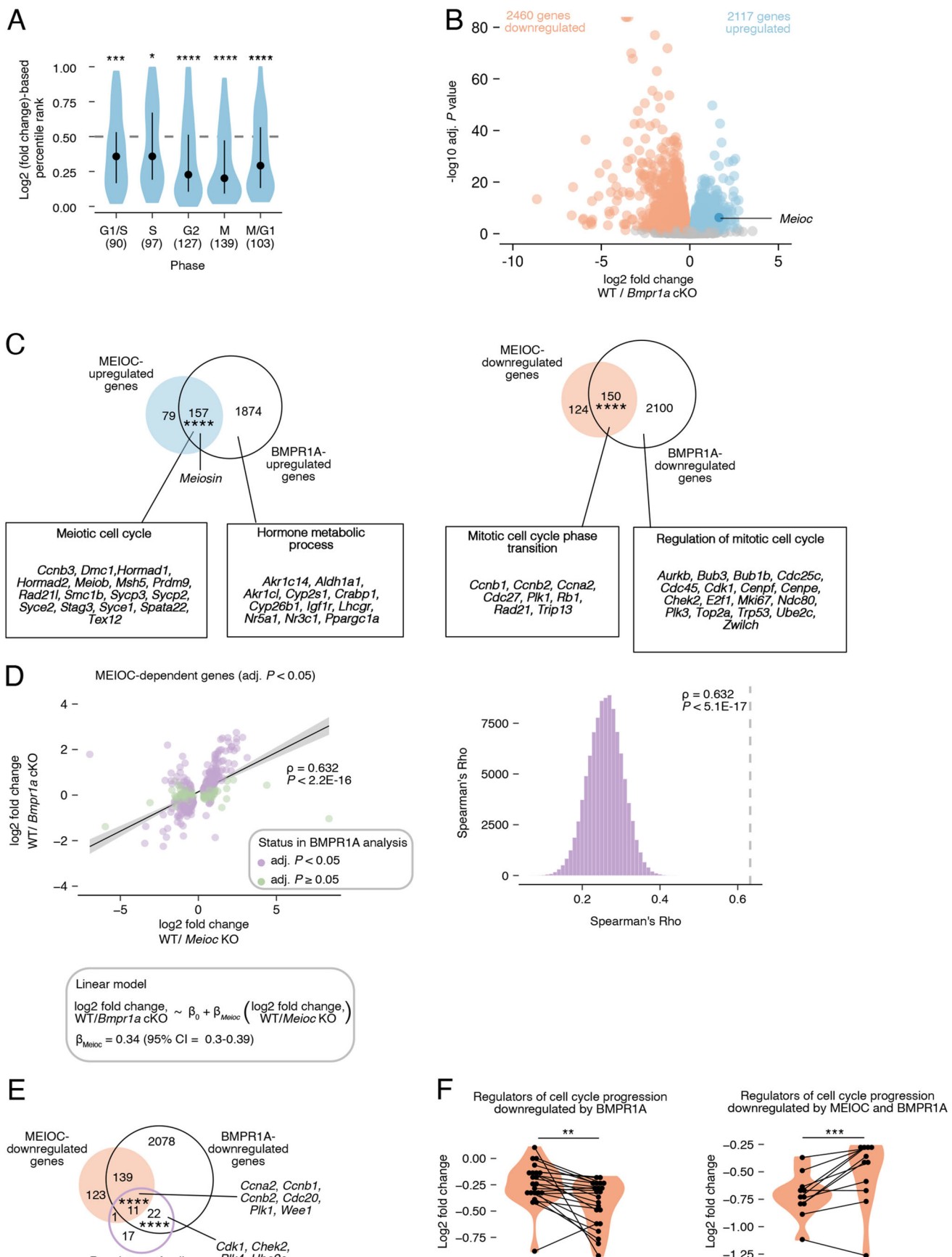

**Fig. 5.** See next page for legend.

**Fig. 5. BMP signaling activates MEIOC to halt mitotic cycling.**
(A) Log$_2$-fold change-based percentile ranks of cell cycle genes for wild-type versus *Bmpr1a* conditional knockout whole ovary RNA-seq data. Dot represents median; whiskers represent the interquartile range. (B) Volcano plot of differential expression results from re-analysis of RNA-seq data generated from E14.5 wild-type and *Bmpr1a* conditional knockout whole ovaries. (C) Venn diagrams comparing gene sets. Boxes highlight select Gene Ontology terms and associated genes enriched within each gene set. *Meiosin* is not yet annotated as a 'meiotic cell cycle' gene and is highlighted separately. (D) Left: correlation between BMPR1A and MEIOC RNA-seq analyses of whole ovaries at E14.5 for MEIOC-dependent genes (adjusted *P*<0.05). *P*-value represents the probability that Spearman rho does not equal 0. β$_{Meioc}$ represents slope in the linear model. Right: distribution of correlations for gene sets obtained by random sampling of genes expressed in BMPR1A and MEIOC RNA-seq analyses. *P*-value represents that probability of obtaining an equal or larger correlation by random sampling. (E) Venn diagrams comparing BMPR1a-downregulated and MEIOC-downregulated genes to a curated set of factors that regulate cell cycle progression. (F) MEIOC- and BMPR1A-dependent log$_2$-fold changes for regulators of cell cycle progression that are downregulated by both MEIOC and BMPR1A, and by BMPR1A only. *adj. *P*<0.05; **adj. *P*<0.01; ***adj. *P*<0.001; ****adj. *P*<0.0001. See Table S6 for statistical details.

is less than 1. Indeed, the linear model yielded a slope (β$_{Meioc}$) of 0.34 (95% confidence interval: 0.30-0.39; Fig. 5D), which indicates that *Bmpr1a* conditional knockout model exhibits reduced MEIOC activity. These results support the model that BMP signaling upregulates MEIOC to yield changes in the transcriptome.

Finally, we examined the biological processes enriched in gene sets regulated by MEIOC and BMP signaling. Genes upregulated by MEIOC and BMPR1A were enriched for Gene Ontology terms associated with the meiotic cell cycle (Fig. 5C). In addition, both MEIOC and BMPR1A upregulated *Meiosin*. Given that BMP signaling acts upstream of MEIOC, we propose that the effects of BMP signaling on meiotic entry occur through MEIOC. Genes upregulated by BMPR1A only were enriched for Gene Ontology terms related to hormone metabolism. These genes include *Crabp1*, whose protein binds to and affects the availability of retinoic acid, and *Aldh1a1*, which encodes a retinoic acid-synthesizing enzyme that supports timely initiation of meiosis in the fetal ovary (Bowles et al., 2016; Isoherranen and Wen, 2025). This suggests that BMP signaling may affect how oogenic cells respond to hormones, including retinoic acid. In addition, the impact of BMP on oogenic development may induce changes in hormone metabolism in the somatic ovary.

Genes downregulated by both MEIOC and BMPR1A, as well as BMPR1A only ,were enriched for Gene Ontology terms associated with mitotic cycling (Fig. 5C). Notably, *Ccna2* was downregulated by both factors. To better understand the mitotic cell cycle genes downregulated by BMP signaling independently of MEIOC, we refined our cell cycle analysis to consider only genes that control cell cycle progression. Using a curated list of 53 genes previously shown to regulate cell cycle progression (McKinley and Cheeseman, 2017), we examined whether this cell cycle gene set exhibited enrichment among genes downregulated by both MEIOC and BMPR1A; by MEIOC only; and by BMPR1A only. We found that genes downregulated by both MEIOC and BMPR1A, as well as by BMPR1A only, were enriched for these regulators of cell cycle progression (Fig. 5E). Genes downregulated by MEIOC only did not exhibit this enrichment. This suggests that BMP signaling halts mitotic cycling through MEIOC as well as through MEIOC-independent regulation.

To explore this possibility further, we examined the fold changes induced by MEIOC and BMPR1A in genes that regulate cell cycle progression. In the gene set downregulated by both MEIOC and

BMPR1A, MEIOC had a greater impact on the fold change than BMPR1A, indicating that MEIOC is driving these changes (Fig. 5F). In the gene set downregulated by BMPR1A only, BMPR1A had a greater impact than MEIOC, suggesting that BMPR1A may be driving these changes independently of MEIOC. These results support the hypothesis that BMP signaling induces additional changes to support the MEIOC-mediated halt to mitotic cycling. However, as *Meioc* knockout oogonia with unperturbed BMP signaling continue to cycle mitotically, BMP-mediated regulation of the cell cycle independent of MEIOC is not strictly required to prevent additional mitotic cell cycles at meiotic initiation. In total, these data demonstrate that BMP signaling activates *Meioc* gene expression, which consequently halts mitotic cycling and upregulates *Meiosin* expression to promote meiotic entry.

## DISCUSSION

Here, we demonstrate that oogenic cells are molecularly directed by MEIOC to halt their mitotic cycling prior to entering the meiotic cell cycle. Though MEIOC has previously been reported to repress the mitotic program during meiotic prophase I (Soh et al., 2017), our proliferation analysis demonstrates that MEIOC also acts on oogenic cells prior to meiotic entry. We find that MEIOC downregulates protein expression of G1/S phase cyclin CCNA2, likely through its destabilization of *Ccna2* mRNA, in preleptotene oocytes before entry into meiotic prophase I. MEIOC also supports upregulation of *Meiosin*, which encodes part of the heterodimeric transcriptional activator STRA8-MEIOSIN required for entry into meiotic S phase. Via these two functions, MEIOC molecularly links a halt in mitotic cycling to meiotic entry during the mitosis-to-meiosis transition in oogenesis.

During the mitotic cell cycle, CCNA2 supports the G1/S phase transition by phosphorylating RB1 and dissociating complexes between RB1 and LXCXE-motif proteins to expose additional phosphorylation sites (Hinds et al., 1992; Fig. 6; Rosenberg et al., 1995; Zarkowska and Mittnacht, 1997). Once hyperphosphorylated, RB1 is unable to bind to and inhibit transcription factor E2F, which then drives the G1/S phase transition (Harbour and Dean, 2000; Hinds et al., 1992; Johnson et al., 1993; Lundberg and Weinberg, 1998; Qin et al., 1995). Consistent with this activity, higher CCNA2 protein levels in G1 phase accelerates entry into S phase (Resnitzky et al., 1994; Rosenberg et al., 1995). Thus, at the meiotic G1/S phase transition, the downregulation of CCNA2 by MEIOC prevents entry into mitotic S phase (Fig. 6). Meanwhile, the upregulation of *Meiosin* gene expression by MEIOC allows STRA8-MEIOSIN to transcriptionally drive entry into meiotic S phase. To further support entry into S phase, STRA8 sequesters RB1 to activate E2F1-mediated transcription (Fig. 6; Shimada et al., 2023). In the absence of STRA8, MEIOC continues to repress CCNA2, which allows RB1 to continue to repress E2F1. As a result, *Stra8* knockout oogenic cells arrest at the G1/S phase transition without entering meiosis. In the absence of MEIOC, increased levels of CCNA2 and persistent STRA8 repress RB1 and allow E2F1 to drive the G1/S phase transition (Fig. 6). As *Meioc* knockout cells fail to upregulate *Meiosin* gene expression, there are insufficient levels of MEIOSIN protein, and the G1/S phase transition is part of a mitotic cell cycle. However, some *Meioc* knockout oogenic cells do enter meiotic prophase I on a delayed timeline (Abby et al., 2016; Soh et al., 2017). We propose that, upon returning to G1 phase, some *Meioc* knockout oogenic cells are able to stochastically produce a sufficient amount of MEIOSIN protein to drive delayed entry into meiotic S phase. In this way, MEIOC functions alongside STRA8-MEIOSIN to rewire the cell cycle program during the transition from mitosis to meiosis.

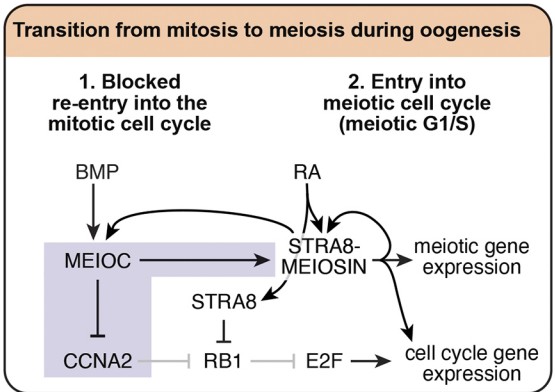

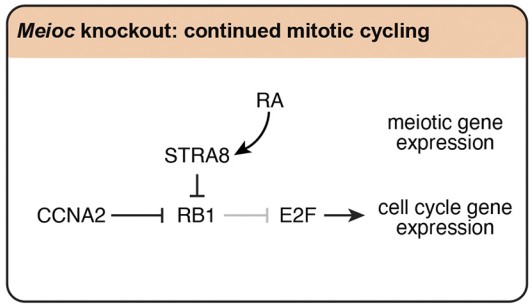

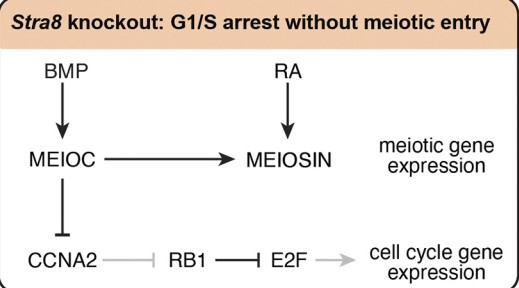

**Fig. 6. MEIOC collaborates with STRA8-MEIOSIN to rewire the cell cycle program as oogenic cells transition from mitosis to meiosis.** Purple highlights the previously unreported regulation discovered in this study. Pointed and blunt black arrows indicate active regulation; pointed and blunt gray arrows indicate regulation that is inactive due to upstream activity. MEIOC inhibits CCNA2 to prevent continued mitotic cycling and activates the STRA8-MEIOSIN transcriptional regulator to drive meiotic entry. In the absence of MEIOC, oogenic cells undergo additional mitotic cycles. Eventually, at least some oogenic cells stochastically express sufficient *Meiosin* levels independently of MEIOC to promote meiotic entry. In the absence of STRA8, oogenic cells arrest at the G1/S phase transition without entering meiosis, although some oogenic cells do slip into an early S phase.

While *Stra8* knockout preleptotene oocytes do not undergo meiotic DNA replication (Baltus et al., 2006), we discovered that they can exhibit some DNA synthesis, identified via incorporation of a thymidine analog. The majority of *Stra8* knockout oocytes arrest at the G1/S phase transition, to which STRA8 molecularly contributes by upregulating expression of G1/S cyclins and DNA replication genes, and interacting with RB1 to sequester it from E2F transcription factors (Baltus et al., 2006; Ishiguro et al., 2020; Kojima et al., 2019; Shimada et al., 2023). The minority of *Stra8* knockout oocytes that exhibit DNA synthesis may have an active G1/S checkpoint but have leaked into an S phase. Alternatively, it is also possible that these cells have successfully completed the G1/S phase transition, but once in S phase, have activated an S checkpoint. Analysis of G1/S and S checkpoints in these cells will be needed to distinguish between these two models. Regardless, the phenotype of *Stra8* knockout oocytes is similar to that in *Stra8* knockout spermatocytes, which also exhibit incorporation of a thymidine analog at the mitosis-to-meiosis transition (Anderson et al., 2008). This DNA replication in *Stra8* knockout spermatocytes and oocytes is not meiotic in nature, as meiotic cohesin REC8 is not loaded onto the chromosomes (Anderson et al., 2008; Baltus et al., 2006). However, it remains unclear whether *Stra8* knockout spermatocytes undergo a partial or complete DNA replication.

The molecular link between BMP signaling and MEIOC remains poorly defined. BMP signaling may activate SMAD transcription factors that directly upregulate *Meioc* expression. It is also possible BMP signaling induces expression of or recruits histone modifiers to remodel chromatin at the *Meioc* locus (Pan et al., 2009; Sun et al., 2009; Yang et al., 2013). The direct and indirect consequences of

BMP signaling on gene expression in fetal oogenic cells requires further investigation.

Our transcriptomic analysis indicates that BMP acts through MEIOC to upregulate *Meiosin* gene expression in oogenic cells (Fig. 6). At the same time, STRA8-MEIOSIN upregulates *Meioc* and *Meiosin* gene expression, and likely also upregulates *Stra8* (Fig. 6; Ishiguro et al., 2020; Kojima et al., 2019; Soh et al., 2015). This positive-feedback loop between MEIOC and STRA8-MEIOSIN is further reinforced by cytoplasmic sharing that occurs through the intercellular bridges of the oogenic cells (Soygur et al., 2021) to establish a robust molecular network that is resistant to perturbations, as disrupted BMP or retinoic acid signaling in the fetal ovary does not prevent meiotic initiation. Instead, genetic ablation of BMP signaling results in delayed meiotic entry (Cheung et al., 2025). Similarly, disrupted retinoic acid signaling via genetic deletion of retinoic acid receptor (RAR) genes or retinaldehyde dehydrogenase genes yields delayed expression of *Stra8* or delayed meiotic entry (Bowles et al., 2016; Chassot et al., 2020; Vernet et al., 2020). Therefore, while neither BMP nor retinoic acid signaling is strictly required for meiotic entry, their combined activities support the positive-feedback loop between MEIOC and STRA8-MEIOSIN to drive the transition from mitosis to meiosis along the timeframe observed during wild-type oogenesis.

We recently reported that, during spermatogenesis, MEIOC destabilizes its mRNA targets, which include transcriptional repressors of *Meiosin* (Pfaltzgraff et al., 2024). Via this activity, MEIOC indirectly activates *Meiosin* gene expression. At the same time, MEIOC downregulates *Stra8* in spermatogenic cells, but MEIOC does not directly bind *Stra8* mRNA, the mechanistic basis

for this regulation remains unclear. As STRA8 and MEIOSIN act as a heterodimer to initiate meiosis, MEIOC enhances the competence of spermatogenic cells to activate the meiotic transcriptional regulator STRA8-MEIOSIN and initiate meiosis in response to retinoic acid. MEIOC may similarly implement this molecular network to regulate *Meiosin* and *Stra8* gene expression during oogenesis. Alternatively, *Meiosin* expression may be developmentally linked to a halt in mitotic cycling, and the regulation of mitotic cycling by MEIOC is sufficient to upregulate *Meiosin* expression. The molecular link between MEIOC and the upregulation of *Meiosin* gene expression warrants further investigation.

In spermatogenic cells, MEIOC collaborates with RNA-binding proteins YTHDC2 and RBM46 to promote the transition from mitosis to meiosis (Pfaltzgraff et al., 2024). MEIOC may similarly act with these binding partners during oogenesis. In support of this hypothesis, *Ythdc2*-null ovaries phenocopy *Meioc*-null ovaries, exhibiting delayed entry into and progression through meiotic prophase as well as premature and abnormal metaphases (Bailey et al., 2017). While *Rbm46*-null postnatal ovaries are devoid of oocytes (Peart et al., 2022), it is not yet known if this phenotype originates during the fetal mitosis-to-meiosis transition. Further work is needed to determine how MEIOC-YTHDC2-RBM46 regulates transcript abundance during oogenesis.

Our discovery that MEIOC halts mitotic cycling during oogenesis raises the possibility that MEIOC may play a similar role during spermatogenesis. Indeed, MEIOC actively shapes the transcriptomes of mitotic spermatogenic cells before the preleptotene stage, and *Meioc* knockout preleptotene spermatocytes diverge from wild-type controls in G1 phase before meiotic S phase. However, further analyses are needed to determine if *Meioc* knockout preleptotene spermatocytes undergo at least one additional mitotic cell cycle before entering meiotic S phase.

In conclusion, in mammalian oogenic cells, BMP-activated MEIOC halts mitotic cycling by downregulating CCNA2 protein expression. This downregulation likely occurs through the destabilization of *Ccna2* mRNA by MEIOC. At the same time, MEIOC promotes the activation *Meiosin* gene expression, which represents a retinoic acid-independent mechanism to support entry into the meiotic cell cycle. These two roles allow MEIOC to molecularly link the halting of mitotic cycling to meiotic entry. This establishes that MEIOC acts alongside STRA8-MEIOSIN to rewire the cell cycle program during the mitosis-to-meiosis transition in oogenic cells.

## MATERIALS AND METHODS
### Ethics statement
All experiments involving mice were performed in accordance with the guidelines of the Massachusetts Institute of Technology (MIT) Division of Comparative Medicine and Cincinnati Children's Hospital Medical Center (CCHMC) Division of Veterinary Services, which are overseen by their respective Institutional Animal Care and Use Committees (IACUC). The animal care programs at MIT/Whitehead Institute and CCHMC are accredited by the Association for Assessment and Accreditation of Laboratory Animal Care, International (AAALAC) and meet or exceed the standards of AAALAC, as detailed in the Guide for the Care and Use of Laboratory Animals. This research was approved by the MIT IACUC (no. 0617-059-20) and CCHMC IACUC (no. 2022-0061).

### Animals
Mice carrying the *Meioc* null allele *Meioc^{tm1.1Dcp}* (RRID: IMSR_JAX:031054; Soh et al., 2017) and the *Stra8* null allele *Stra8^{tm1Dcp}* (RRID:IMSR_JAX:023805; Baltus et al., 2006) were backcrossed to C57BL/6N (B6N) from Taconic Biosciences for at least 10 generations. Additional wild-type B6N mice to generate wild-type

embryos were obtained from the Jackson Laboratory. Timed matings were set up in the evenings and verified by the presence of a vaginal plug in the morning. The midpoint of the dark cycle was denoted at E0. Timed matings between a wild-type male and a wild-type female generated wild-type embryos at E12.5 and E14.5. Timed matings between a male and female, each heterozygous for *Meioc* and *Stra8*, were used to generate E16.5 embryos that were phenotypically wild type (wild-type or heterozygous for *Meioc* and *Stra8*); *Meioc* knockout (homozygous null for *Meioc* and wild-type or heterozygous for *Stra8*); *Stra8* knockout (wild-type or heterozygous for *Meioc* and homozygous null for *Stra8*); and *Meioc Stra8* double knockout (homozygous null for *Meioc* and *Stra8*).

### EdU incorporation
EdU was prepared as a 4 µg/µl solution in PBS. Pregnant mothers at E12.5, E14.5 and E16.5 were intraperitoneally injected with a final dose of 20 µg/g EdU, and embryos were collected 2 h after EdU injection.

### Immunostaining
Embryonic ovaries were fixed in 4% paraformaldehyde (PFA) overnight at 4°C, washed in PBS, embedded in paraffin blocks and 6 µm tissue sections were prepared. Tissue sections were dewaxed in xylene and rehydrated in solutions with decreasing ethanol concentrations. Sections were incubated in antigen retrieval buffer for 15 min in a pressure cooker at 12 PSI and approximately 117°C. For sections to be stained for DDX4, CCNA2 and EdU, Tris-EDTA buffer [10 mM Tris, 1 mM EDTA and 0.05% Tween 20 (pH 9.0)] was used for antigen retrieval. For sections to be stained for SYCP3 and EdU, sodium citrate buffer (10 mM sodium citrate, 0.05% Tween 20, pH 6.0) was used for antigen retrieval. Slides were then cooled for 15 min before washing in PBS.

Tissue sections were blocked in 5% normal donkey serum/PBS for at least 1 h at room temperature and incubated in primary antibody diluted in blocking solution overnight at 4°C. Primary antibodies used were anti-CCNA2 (rabbit polyclonal, Abcam ab181591; 1.344 mg/ml stock; 1:750 working concentration), anti-DDX4 (goat polyclonal, R&D Systems AF2030; 0.2 mg/ml stock; 1:250 working concentration) and anti-SYCP3 (mouse monoclonal, Santa Cruz Biotechnology sc-74569; 0.2 mg/ml stock; 1:200 working concentration). [A subset of slides were stained with anti-REC8 (rabbit polyclonal, Abcam ab192241; 1.344 mg/ml stock; 1:50 working concentration) in place of anti-CCNA2; these slides were used for EdU quantification, but analysis of REC8 immunostaining is not included here.] Slides were washed in PBS and incubated in secondary antibody in a dark chamber for 1 h at room temperature. Secondary antibodies anti-goat Alexa Fluor 488 (ThermoFisher A-11055, 2.0 mg/ml stock), anti-rabbit Alexa Fluor 568 (Abcam ab175692, 2.0 mg/ml) and anti-mouse Alexa Fluor 488 (ThermoFisher A-21202, 2.0 mg/ml stock) were used at a working concentration of 1:250 in blocking buffer. Slides were washed in PBS and EdU was labeled with Alexa Fluor 647 using the Click-iT EdU Imaging Kit (ThermoFisher Scientific C10340), following the manufacturer's instructions. The slides were then mounted with ProLong Glass Antifade Mountant (ThermoFisher Scientific P36984) and 1.5H coverslips (Thorlabs, CG15KH).

### Microscopy and image analysis
Slides stained for DDX4 and EdU were imaged on a Nikon Yokogawa Sora W1 confocal spinning disk microscope with Hamamatsu FusionBT SCMOS camera in the spinning disk fluorescence modality using a 50 µm pinhole with a Nikon CFI Apochromat LWD Lambda S 20× water immersion objective (numerical aperture=0.95; refractive index=1.333) using NIS Elements software [NIS-Elements 5.42.06 (Build 1821) LO, 64bit, Nikon]. Laser excitations of 405 nm, 488 nm, 561 nm and 640 nm were used for DAPI, Alexa Fluor 488 (GFP), Alexa Fluor 568 (RFP) and Alexa Fluor 647 (Cy5), respectively. Emission signals were captured through corresponding emission filters at 455/50 nm, 525/36 nm, 605/52 nm and 705/72 nm. 10-20 z-stack images were acquired with a step size of 0.6 µm through 6 µm sections and converted into maximum intensity projection (MIP) images, which were used for image analysis.

Slides stained for SYCP3 and EdU were imaged on a Nikon A1+ confocal laser scanning microscope and camera using a 26.82 µm pinhole with a

Nikon CFI Plan Apochromat LWD Lambda S 40× water immersion objective (numerical aperture=1.15; refractive index=1.333) using NIS Elements software [NIS-Elements 5.42.01 (Build 1793) LO, 64bit, Nikon]. Laser excitations of 405 nm, 488 nm and 640 nm were used for DAPI, Alexa Fluor 488 (GFP) and Alexa Fluor 647 (Cy5), respectively. Emission signals were captured through corresponding emission filters at 452/45 nm, 525/50 nm and 705/72 nm. 3-5 $z$-stack images were acquired with a step size of 2.5 μm through 6 μm sections and converted into maximum intensity projection (MIP) images, which were used for image analysis.

For slides stained for DDX4 and EdU, cell counts and CCNA2 intensity were quantified using NIS-Elements NIS.ai software. First, an artificial intelligence module "Segment objects AI" was trained to segment oogenic cells in ovarian tissue sections using cytoplasmic DDX4 in the GFP channel. Then, segmented cells were binned as positive or negative for EdU in the Cy5 channel and positive or negative for CCNA2 in the RFP channel. For CCNA2-positive cells, mean object intensity in the RFP channel was also recorded.

For slides stained for SYCP3 and EdU, SYCP3 signal was used to identify oocytes in zygotene and pachytene stages by eye, and these cells were then designated as positive or negative for EdU. Images have been deposited in the BioImage Archive under accession number S-BIAD2080. The trained artificial intelligence module and analysis pipeline are available at https://github.com/Mikedis-Lab/2025_fetal_oocytes_image_analysis.

### Statistical analysis of image data

Percentage of DDX4-positive oogenic cells that were EdU positive were first compared using a Chi-squared test of homogeneity via R function *chisq.test*, which indicated that the data from four genotypes did not come from the same distribution ($P<0.05$). To identify different data distributions between genotypes, post-hoc Chi-squared tests of homogeneity were used, and $P$-values were adjusted for multiple hypothesis testing via the Benjamin-Hochberg correction using the *p.adjust* function in R.

To demonstrate that SYCP3-positive oocytes at zygotene and pachytene stages did not incorporate EdU during double-strand break repair in wild-type ovaries at E16.5, a power calculation was used to determine the appropriate sample size for the zygotene/pachytene population for a power of 0.8 and significance level of 0.05. The power calculation for a two-sample comparison of proportions with unequal sample sizes was carried out using the *power_Binomial* function from R package PASSED. Given that the analysis of DDX4-positive oocytes ($n$=7703) identified 4.5% as EdU-positive in wild-type ovaries at E16.5, and that oocytes in zygotene and pachytene stages were predicted to be negative for EdU, at least 90 SYCP3-positive oocytes at zygotene and pachytene stages would need be quantified for EdU status. All zygotene and pachytene oocytes in all ovarian sections stained for SYCP3 and EdU were ultimately quantified for EdU status, resulting in a total sample size of 317 oocytes from three embryos. The frequency of EdU in zygotene and pachytene oocytes was compared to the frequency of EdU in DDX4-positive oocytes using an exact binomial test via the *bionom.test* function in R.

### Re-analysis of 10x Genomics scRNA-seq data

10x Genomics scRNA-seq data (NCBI GEO GSE130212 from Zhao et al., 2020) from E12.5 and E14.5 fetal ovarian germ cells, isolated via sorting for the *Pou5f1*:EGFP reporter, was re-analyzed. Alignment, filtering, barcode counting and UMI counting were carried out using the *count* function in Cell Ranger v.7.0.1 (options: –include-introns="false") using Cell Ranger's mm10-2020-A reference package (i.e. the GRCm38/mm10 mouse genome assembly with GENCODE vM23/Ensembl 98 annotation). Using Seurat v.4.1.1 (Stuart et al., 2019), cells were filtered for less than 10% mitochondrial reads, more than 1000 detected features, and a doublet score (calculated via the *bcds* function in scds v.1.6.0) of less than 0.4. UMI counts for protein-coding genes from all samples were integrated and clusters identified using the first 10 dimensions.

Cell types were assigned to clusters based on cell type-enriched gene expression in combination with cell cycle phase using the *CellCycleScoring* function in Seurat, as previously described for the mitosis-to-meiosis transition during spermatogenesis (Pfaltzgraff et al., 2024). E12.5 cells were further used to distinguish E14.5 mitotic oogenic cells from preleptotene

and later stages: as E12.5 oogenic cells are all mitotic, E14.5 oogenic cells that clustered with the E12.5 data were designated as mitotic, while E14.5 oogenic cells that clustered separately from the E12.5 data were confirmed as preleptotene or later stages. Clusters were merged as needed, cells were filtered for E14.5 only, and a somatic cell cluster was removed. Dot plots were generated using Seurat's *DotPlot* function.

To identify genes enriched and depleted in a single oogenic cell clusters, relative to all other oogenic clusters, Seurat's *FindAllMarkers* function (options: min.pct=0.1, logfc.threshold=0.05) was used. Depletion of *Mki67* expression in pL G1 cluster and enrichment of replication-dependent histone expression in pL eS and lS clusters further confirmed cell cycle designations independent of the gene:cell cycle phase pairings used by Seurat's *CellCycleScoring* function (Fig. S2E,F).

For differential expression analysis between two sequential oogenic cell clusters, $log_2$-fold change was defined as the later cluster over earlier cluster, such that the value reflects transcript abundance changes as the oogenic cells transition from the earlier to later stage. Differential expression was tested for genes that were detected in at least 10% of either cluster's population and without a minimum absolute $log_2$-fold change using Seurat's *FindMarkers* function (options: min.pct=0.1, logfc.threshold=0); $P$-values were adjusted for multiple hypothesis testing of tested genes across five pairs of sequential clusters via the Bonferroni correction using the *p.adjust* function in R.

To identify genes differentially expressed between mitotic and preleptotene oogenic cells in G1 phase, Mit and pL G1 clusters were filtered for those cells designated as G1 phase. Differential expression for filtered pL G1 versus Mit clusters was then tested as described above.

To identify genes differentially expressed between mitotic and preleptotene oogenic cells in S phase, Mit, pL eS and pL lS clusters were filtered for those cells designated as S phase. Differential expression for filtered pL eS/lS versus Mit clusters was then tested as described above.

### Re-analysis of STRA8, MEIOC and BMPR1A bulk RNA-seq datasets

Whole ovary bulk RNA-seq from wild-type and *Stra8* knockout samples (NCBI GEO GSE70361 from Soh et al., 2015), wild-type and *Meioc* knockout samples (NCBI GEO GSE90702 from Soh et al., 2017), as well as wild-type and *Bmpr1a* conditional knockout samples (NCBI GEO GSE268565; Cheung et al., 2025) were re-analyzed. Reads were quality trimmed using cutadapt v1.8 (options: -q 30 –minimum-length 20 -b AGATCGGAAGAGC). Expression levels of all transcripts in the mouse Gencode Basic vM15 gene annotation, with *Gcna* manually added, were estimated using kallisto v0.44.0 (Bray et al., 2016) with sequence-bias correction (–bias). Quantified transcripts were filtered for protein-coding genes, transcript-level estimated counts and transcripts per million (TPM) values were summed to the gene level, and TPMs were renormalized to transcript-per-million units. Within each dataset, genes were filtered for a minimum TPM of 1 in at least three of six samples (in MEIOC and STRA8 datasets) or at least three of eight samples (in BMPR1A dataset). Read counts from kallisto were rounded to the nearest integer, and each dataset was individually supplied to DESeq2 v1.26.0 for analysis of differential expression/abundance. The datasets were then merged based on gene name.

Cell cycle analysis of bulk RNA-seq data was implemented as previously described (Pfaltzgraff et al., 2024). Briefly, a curated list (Hsiao et al., 2020) of human cell cycle genes whose expression is associated with specific cell cycle phases was converted to one-to-one mouse orthologs. The percentile ranks of genes, based on $log_2$-fold change [wild type/knockout (WT/KO)], for each cell cycle phase were compared to median percentile rank for all expressed genes (0.5) via a two-sided Wilcoxon singed rank test with Benjamini-Hochberg correction for multiple hypothesis testing, as implemented by the *wilcox.test* and *p.adjust* functions in R.

MEIOC-upregulated genes were defined as $log_2$ fold change WT/*Meioc* KO>0 and adjusted $P<0.05$ and MEIOC-downregulated genes were defined as $log_2$ fold change WT/*Meioc* KO<0 and adjusted $P<0.05$. STRA8-upregulated genes were defined as $log_2$-fold change WT/*Stra8* KO>0 and adjusted $P<0.05$ and STRA8-downregulated genes were defined as $log_2$-fold change WT/*Stra8* KO<0 and adjusted $P<0.05$. BMPR1A-upregulated genes were defined as $log_2$-fold change WT/*Bmpr1a* cKO>0 and adjusted $P<0.05$ and BMPR1A-downregulated genes were defined as $log_2$ fold change

WT/*Bmpr1a* cKO<0 and adjusted $P<0.05$. A curated list of factors that regulate cell cycle progression was obtained from McKinley and Cheeseman (2017). To determine whether MEIOC-upregulated genes were enriched for STRA8-upregulated genes; whether MEIOC-downregulated genes were enriched for STRA8-downregulated genes; whether MEIOC-upregulated genes were enriched for BMPR1A-upregulated genes; whether MEIOC-downregulated genes were enriched for BMPR1A-downregulated genes; whether MEIOC and BMPR1A-downregulated genes are enriched for regulators of cell cycle progression; whether genes downregulated by BMPR1A but not MEIOC are enriched for regulators of cell cycle progression; and whether genes downregulated by MEIOC but not BMPR1A are enriched for regulators of cell cycle progression, these gene lists were statistically compared using a one-tailed hypergeometric test via the *phyper* function with *lower.tail=F* in R. The Benjamini-Hochberg correction was used to account for multiple hypothesis testing, as implemented by the *p.adjust* function in R.

The *cor.test* function in R (options: method='‘spearman’) was used to calculate Spearman rank correlation rho (ρ) between the MEIOC and STRA8 bulk RNA-seq datasets' $log_2$-fold change for STRA8-dependent genes only (adjusted $P<0.05$) and between MEIOC and BMPR1A bulk RNA-eq datasets' $log_2$-fold change for MEIOC-dependent genes only (adjusted $P<0.05$). In the graphs of these data, the small number of extreme data points that fell outside the axes of the graph are displayed as 0.1 outside the axis limit. For the bootstrapping analysis, Spearman rank correlation rhos were calculated for 100,000 gene sets randomly generated from all expressed genes in the bulk RNA-seq datasets. The number of sampled genes was equal to the number of STRA8-dependent genes (for the MEIOC versus STRA8 analysis) or MEIOC-dependent genes (for the BMPR1A versus MEIOC analysis). The Spearman rhos from random sampling were used to calculate a Z-score for the Spearman rho of the STRA8-dependent genes in MEIOC versus STRA8 analysis, or the MEIOC-dependent genes in the BMPR1A versus MEIOC analysis. *P*-value was calculated from the Z-score using the *pnorm* function in R (options: lower.tail=FALSE, log.p=TRUE).

The linear model between the MEIOC-dependent and STRA8-dependent $log_2$ fold changes for STRA8-dependent genes only (adj. $P<0.05$) was calculated using the *lm* function in R using the following equation:

$$log2 \ fold \ change \ WT/Meioc \ KO \sim \beta_0$$
$$+ \beta_{Stra8}(log2 \ fold \ change \ WT/Stra8 \ KO).$$

In this equation, $\beta_0$ represents the intercept and $\beta_{Stra8}$ represents the coefficient of the effect of STRA8 on transcript abundance.

Similarly, the linear model between the BMPR1A-dependent and MEIOC-dependent $log_2$-fold changes for MEIOC-dependent genes only (adj. $P<0.05$) was calculated using the *lm* function in R using the following equation:

$$log2 \ fold \ change \ WT/Bmpr1a \ cKO$$
$$\sim \ \beta_0 + \beta_{Meioc}(log2 \ fold \ change \ WT/Meioc \ KO).$$

In this equation, $\beta_0$ represents the intercept and $\beta_{Meioc}$ represents the coefficient of the effect of MEIOC on transcript abundance.

### Functional analysis of gene sets from scRNA-seq and bulk RNA-seq data

For sequential pairwise comparisons between scRNA-seq clusters, upregulated ($log_2$ fold change>0 and adjusted $P$-value<0.05) and downregulated genes ($log_2$ fold change<0 and adjusted $P$-value<0.05) were separately tested for enrichment of Gene Ontology Biological Processes gene lists relative to a background list of all expressed genes (i.e. genes expressed in at least 10% of cells within either of the two compared clusters). For bulk RNA-seq datasets, sets of upregulated ($log_2$ fold change>0 and adjusted $P$-value<0.05) and downregulated ($log_2$ fold change<0 and adjusted $P$-value<0.05) genes were similarly tested against a background of all expressed genes. Gene Ontology analyses were carried out using the *enrichGO* function (options: OrgDb="org.Mm.eg.db", ont="BP", pvalueCutoff=0.05, readable=T, pAdjustMethod="BH") in R package clusterProfiler v3.0.4.

STRA8-activated genes in embryonic oocytes were defined as genes that were upregulated by STRA8 ($log_2$ fold change>0 and adjusted $P$-value<0.05) in bulk RNA-seq analysis of E14.5 ovaries and bound by STRA8 at their promoters, based on ChIP-seq data from preleptotene spermatocytes (defined by Kojima et al., 2019). Enrichments of STRA8-activated genes among STRA8-upregulated and MEIOC-upregulated genes from bulk RNA-seq analysis of E14.5 ovaries were statistically compared to those in all expressed genes using a one-tailed hypergeometric test via the *phyper* function with *lower.tail=F* in R. The Benjamini-Hochberg correction was used to account for multiple hypothesis testing, as implemented by the *p.adjust* function in R.

To define STRA8-activated genes among meiosis-associated genes that are upregulated by MEIOC and STRA8, meiosis-associated genes were identified using the following enriched Gene Ontology terms: GO:0007127 meiosis I; GO:0007129 homologous chromosome pairing at meiosis; GO:0045132 meiotic chromosome segregation; GO:0045143 homologous chromosome segregation; GO:0051321 meiotic cell cycle; GO:0061982, meiosis I cell cycle process; GO:007019 chromosome organization involved in meiotic cell cycle; GO:0140013 meiotic nuclear division; and GO:1903046 meiotic cell cycle process. Genes were aggregated and filtered to generate a unique gene list of meiosis-associated genes that are upregulated by MEIOC and STRA8, STRA8 only, and MEIOC only. Genes activated by STRA8 were annotated as those with a STRA8-bound promoter and upregulated by STRA8 in preleptotene spermatocytes, as defined by Kojima et al. (2019).

### Acknowledgements
We thank David C. Page for his support when this project was in its initial stages. We thank Mina Kojima, Holly Christensen and Peter Nicholls for technical help; Amanda Barbosa and Sarah McLeod for assistance with microscopy and image analysis; and Jordana Bloom for valuable feedback. Fluorescent microscopy was carried out at Cincinnati Children's Bio-Imaging and Analysis Facility (RRID: SCR_022628).

### Competing interests
The authors declare no competing or financial interests.

### Author contributions
Conceptualization: J.T.N., M.M.M.; Data curation: E.G.U., S.M.W., M.M.M.; Formal analysis: E.G.U., J.T.N., S.M.W., M.M.M.; Funding acquisition: M.M.M.; Investigation: E.G.U., J.T.N., N.G.P., M.M.M.; Methodology: E.G.U., J.T.N., S.M.W., M.M.M.; Project administration: M.M.M.; Software: E.G.U., J.T.N., M.K., M.M.M.; Supervision: M.M.M.; Validation: E.G.U.; Visualization: E.G.U., J.T.N., S.M.W., M.M.M.; Writing – original draft: M.M.M.; Writing – review & editing: E.G.U., J.T.N., N.G.P., M.K., M.M.M.

### Funding
This work was supported by the Eunice Kennedy Shriver National Institute of Child Health and Human Development of the National Institutes of Health under award numbers K99HD097285 (M.M.M.) and R00HD097285 (M.M.M.). Open Access funding provided by the University of Cincinnati. Deposited in PMC for immediate release.

### Data and resource availability
Images have been deposited in the BioImage Archive under accession number S-BIAD2080. The trained artificial intelligence module and analysis pipeline are available at https://github.com/Mikedis-Lab/2025_fetal_oocytes_image_analysis. All other relevant data and details of resources can be found within the article and its supplementary information.

### The people behind the papers
This article has an associated 'The people behind the papers' interview with some of the authors.

### Peer review history
The peer review history is available online at https://journals.biologists.com/dev/lookup/doi/10.1242/dev.205037.reviewer-comments.pdf

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
