## [Peer Review File · Development (Cambridge, England)]

MEIOC prevents continued mitotic cycling and promotes meiotic entry during mouse oogenesis

Esther G. Ushuhuda, Jenniluyun T. Nguyen, Natalie G. Pfaltzgraff, Shelbie M. Wenner, Matthew Kofron and Maria M. Mikedis
DOI: 10.1242/dev.205037

Editor: Swathi Arur

Review timeline

Original submission:	16 June 2025
Editorial decision:	16 July 2025
First revision received:	22 October 2025
Accepted:	10 November 2025

Original submission

First decision letter

MS ID#: dev.205037

MS TITLE: MEIOC prevents continued mitotic cycling and promotes meiotic entry during mouse oogenesis

AUTHORS: Esther G. Ushuhuda, Jenniluyun T. Nguyen, Natalie G. Pfaltzgraff, Shelbie M. Wenner, Matthew Kofron and Maria M. Mikedis

Dear Dr Mikedis,

I have now received all the referees' reports on the above manuscript, and have reached a decision. The referees' comments are appended below, or you can access them online: please go to:

As you will see, the referees express considerable interest in your work, but offer significant recommendations to greatly improve the clarity, and rigor of the work. Mainly, as noted by Reviewer 2 and 3 the manuscript makes very strong claims on the regulation of MEIOSIN at the transcriptional level by BMP signaling. However, the data presented is largely correlative. Thus, it would be important to either present some direct evidence of this transcriptional regulation, by binding etc, or greatly soften the claims. All three reviewers point out similar statements throughout the manuscript which are stronger than the evidence presented. I recommend greatly contextualizing the findings in the light only of the evidence provided. Reviewer 1 makes an excellent suggestion on the distinctions of meiotic S phase eluded to by the authors. I recommend that the authors carefully review the manuscript for consistency in definitions. Overall, all three reviewers make significant recommendations both textual or some that may require further experimentation. If you are able to revise the manuscript along the lines suggested, which may involve further experiments, I will be happy receive a revised version of the manuscript. Your revised paper will be re-reviewed by one or more of the original referees, and acceptance of your manuscript will depend on your addressing satisfactorily the reviewers' major concerns. Please also note that Development will normally permit only one round of major revision.

Please attend to all of the reviewers' comments and ensure that you clearly highlight all changes made in the revised manuscript. Please avoid using 'Tracked changes' in Word files as these are lost in PDF conversion. I should be grateful if you would also provide a point-by-point response detailing

how you have dealt with the points raised by the reviewers in the 'Response to Reviewers' box. If you do not agree with any of their criticisms or suggestions please explain clearly why this is so.

Reviewer 1

Advance summary and potential significance to field

In this manuscript, the authors evaluate pathways that control meiotic entry in the mammalian female germ line. The knowledge about the biology of meiotic entry, especially in females where entry happens during fetal development, is thin. I find this manuscript thoughtful in their approach at how to make discoveries largely using existing datasets and then validations in mouse knockout models. They conclude that Meioc prevents mitotic cycling by downregulating Ccna2 protein. They also connect Meioc signaling with the Stra8-Meiosin pathway and demonstrate that Bmp signaling is upstream of Meioc. Overall, I find the manuscript interesting. I have some questions, points of clarity and suggestions for the authors to consider in a revised manuscript.

1- In the discussion of Figure 1, the authors indicate that their findings of reduced Edu labeling conflicts with another paper (Baltus), but this is likely do to different assays used. Later the authors mention that the S phase that does occur, is not meiotic S because Rec8 is not loaded (from Anderson). To me, this seems like an important distinction to be made earlier when highlighting differences and supporting their conclusion that you need Stra8 for meiotic S phase. Otherwise, just by taking their figure and the Baltus data into consideration, one could say that it is dispensable for S phase, because they can still replicate, just not as effectively (line 101).

2- I found text in Line 112 confusing. Do the authors mean that the Stra8 KO cells are mitotically cycling? How do they know that this happening by their rationale of the cell cycle block? I think the rationale of thought is not obvious (or direct).

3- In the y-axis of Edu labeling of graphs, the authors use "Edu fraction." Is this the same as percent of cells that are Edu Positive?

4- To me, the term preleptotene is confusing. This is because a prophase substage is thrown into the cell cycle terms. I realize that this is field standard, but perhaps it would be worth defining preleptotene? Are these cells that are in S phase? Are these cells that are in prophase? Or are these cells that are in some sort of undefined transition state?

5- In figure 3D the authors state that the small difference in RNA expression is consistent with a role in RNA destabilization. I'm curious why the authors conclude that the mechanism is at the RNA level and not just at the protein level? There are many reasons why a protein expression could be altered. Why do the authors choose this mechanism? Because the mechanism is completely speculative, the authors could expand on this.

6- I found the use of full sentences in parenthesis distracting.

7- It seems like this mechanism is similar to how male meiotic entry is also regulated. This should be discussed in the discussion (or if anything is different).

8- In studies where the authors counted edu or other labeling, a total number of animals and cells should be included in the legends.

9- In Edu incorporation, the authors indicate collecting embryos. At these stages, aren't they a fetus?

10- Line 534- I don't usually see the word "cover slipped" as a verb- lab jargon?

Reviewer 2

Advance summary and potential significance to field

The authors present data and re-analysis of genomic datasets to make conclusions regarding the mechanism by which mouse oogenesis change from the mitotic DNA replication program to premeiotic DNA replication, and subsequently enter the meiotic gene expression program. While there is substantial data from the male side (spermatogonia>spermatocyte), oogenesis is harder to access and obviously has much different biology, so the question is relevant, important, and of substantial interest.

The experimental parts of this paper are essentially EdU labeling experiments of germ cells from embryonic ovaries of *Stra8* and *Meioc* knockouts, enabling an assessment of DNA replication activities. The bulk are from re-analysis of single cell and bulk RNA-seq datasets that were generated elsewhere, with a focus on transcript levels related to the cell cycle. Combining the bioinformatics and EdU experiments, the authors come up with a model of the regulatory circuitry summarized in Fig. 6 and summarized by the paper's title regarding the key role of MEIOC.

Overall, the topic is important and the conclusions drawn seem reasonable, albeit many aspects are correlative, not definitive. Gaining definitive molecular proof of the relationships in Fig. 6 would require extensive targeted experimentation, so in the meantime, I feel that this paper contributes a valuable conceptual framework.

Comments for the author

MAJOR POINTS

- with exception of assaying protein levels of *CCNA2*, no assessments of protein levels of cyclins were done nor were CDKs phosphorylation states assessed. Thus, nearly all the conclusions of cell cycle stages at different developmental time points were based on scRNA-seq data. Not sure how easy it would be to do Westerns on limiting quantities of sample.
- lines 273-276 "...we found that the *STRA8*-dependent genes (adjusted $P < 0.05$) exhibited significantly correlated effects in the MEIOC and *STRA8* datasets (Figure 4C; Table S5). This confirms that the loss of MEIOC disrupts *STRA8*-*MEIOSIN* transcriptional activity." Correlations are not confirmation. Also, transcript abundance is not tantamount to transcriptional activity; one would have to measure transcription directly via methods such as run-on sequencing. In the absence of such data (I don't know if it exists), there should be ample caveats stated. Also, the section on the role of BMP signaling on MEIOC is also correlative, not definitive. At some point, I imagine in vitro gametogenesis could be useful to validate gene regulation questions.
- unlike studies of male meiosis, the authors do not perform analyses of direct binding by *STRA8* to genes they conclude that are directly regulated by *STRA8*/*MEIOSIN*, or (for example), look at ATAC-seq data in WT vs mutants.

MINOR POINTS

- Lines 88-90. For the non-expert reader, should specify that "oogenic cells" are defined as those that are *DDX4*/*MVH*-positive. It wouldn't hurt to say a few words about this protein's temporal expression pattern. Similarly for *Pou5f1* (relevant for the section beginning on line 138).
- Lines 98-101. It isn't clear to me how the conclusion is reached that some oogenic cells "initiate some DNA synthesis but fail to complete meiotic DNA replication." Perhaps explain more clearly (rather than just pointing to papers) that pre-meiotic DNA replication can stop in early S-phase once started (in the absence of checkpoint activation). Because the data in Fig. 1 is scoring cells as being positive or negative, is it possible that weaker staining reflects EdU from DNA repair? Similarly, on lines 108-109, it isn't clear how cells would be doing any DNA replication at the G1/S phase transition. Are the authors hypothesizing that such cells activate some licensed origins but never reach the restriction point?
- line 277 "As MEIOC exhibits reduced, but not complete loss of, Meiosin gene expression..." Do the authors mean MEIOC absence?

- Fig. 6 Suggest that in the Stra8 KO condition, put a slash through "cell cycle gene expression" since E2F will be inhibited by Rb1 in this scenario, if I understand correctly.

Reviewer 3

Advance summary and potential significance to field

Proper initiation of meiosis requires both the programmatic switch from mitotic to meiotic cell division and the activation of meiosis-specific genes. In previous studies, the authors demonstrated that MEIOC plays a crucial role in post-transcriptional regulation during male meiotic initiation. In this study, they further show that MEIOC is also required for the initiation of the mitosis-to-meiosis transition in oocytes. In addition to previously reported phenomena observed in males, the authors uncover female-specific features, such as the involvement of BMP signaling in regulating Meioc expression. These findings provide valuable insights and contribute significantly to the field. The data presented are generally consistent with the genetic network model proposed by the authors. However, the main concern with this manuscript lies in the limited ability to infer molecular mechanisms based on the bulk RNA-sequencing data, which were obtained from a single time point. Incorporating data that offer mechanistic insights—such as time-course analyses, single-cell transcriptomics, or functional assays—would substantially enhance the impact of this study.

Comments for the author

Major points to address:

Lines 248-314:

The authors discuss the effect of MEIOC on STRA8-MEIOSIN function based on bulk RNA-seq data collected at a single developmental time point. However, as the authors themselves acknowledge, Meioc-KO oocytes exhibit delayed meiotic initiation, implying a shift in developmental timing. Thus, it remains unclear whether the observed changes in Meiosin expression reflect a direct regulatory effect by MEIOC, or are instead a consequence of this developmental delay. The current data do not allow the authors to definitively conclude that MEIOC directly regulates STRA8-MEIOSIN activity. This issue should be carefully reconsidered and appropriately qualified in the manuscript.

Lines 316-392:

The authors discuss the regulation of Meioc by BMP signaling. While there is little doubt that BMP signaling influences Meioc expression, it remains unresolved whether this regulation is direct or mediated indirectly via BMP-induced changes in cellular context. Clarifying this point—or at least explicitly discussing the possibilities—would strengthen the interpretation and is especially relevant when considering sex-specific differences in the regulation of Meioc and meiotic initiation.

Lines 933-936:

The labeling "Meiosis-associated genes upregulated by MEIOC and STRA8" in Fig4D is not consistent with "Fraction of meiosis-associated genes upregulated by MEIOC and BMPR1A" in the legend.

First revision

Author response to reviewers' comments

We thank the three reviewers for their time, as their thorough readings and thoughtful comments have helped us to greatly improve the manuscript's analyses and clarity. In response to their comments, we have added new analyses and made numerous revisions to text and figures. We detail our revisions in response to the comments below. Within the manuscripts, revisions are highlighted in yellow.

Reviewer 1

In this manuscript, the authors evaluate pathways that control meiotic entry in the mammalian female germ line. The knowledge about the biology of meiotic entry, especially in females where entry happens during fetal development, is thin. I find this manuscript thoughtful in their approach at how to make discoveries largely using existing datasets and then validations in mouse knockout models. They conclude that Meioc prevents mitotic cycling by downregulating Ccna2 protein. They also connect Meioc signaling with the Stra8-Meiosin pathway and demonstrate that Bmp signaling is upstream of Meioc. Overall, I find the manuscript interesting. I have some questions, points of clarity and suggestions for the authors to consider in a revised manuscript.

1- In the discussion of Figure 1, the authors indicate that their findings of reduced EdU labeling conflicts with another paper (Baltus), but this is likely do to different assays used. Later the authors mention that the S phase that does occur, is not meiotic S because Rec8 is not loaded (from Anderson). To me, this seems like an important distinction to be made earlier when highlighting differences and supporting their conclusion that you need Stra8 for meiotic S phase. Otherwise, just by taking their figure and the Baltus data into consideration, one could say that it is dispensable for S phase, because they can still replicate, just not as effectively (line 101).

We have added text to clarify that the *Stra8* knockout does not exhibit meiotic DNA replication because REC8 is not loaded onto the chromosomes (lines 97-98).

We also clarify that our EdU results do not conflict with Baltus et al.; instead, they provide a more nuanced characterization of the *Stra8* knockout phenotype in oogenic cells (lines 99-104).

2- I found text in Line 112 confusing. Do the authors mean that the Stra8 KO cells are mitotically cycling? How do they know that this happening by their rationale of the cell cycle block? I think the rationale of thought is not obvious (or direct).

We have modified the text to clarify our reasoning (lines 104-109).

3- In the y-axis of Edu labeling of graphs, the authors use "Edu fraction." Is this the same as percent of cells that are Edu Positive?

We have updated the y-axis of Figure 1C to reflect the "Percent of DDX4-positive cells that are EdU-positive."

4- To me, the term preleptotene is confusing. This is because a prophase substage is thrown into the cell cycle terms. I realize that this is field standard, but perhaps it would be worth defining preleptotene? Are these cells that are in S phase? Are these cells that are in prophase? Or are these cells that are in some sort of undefined transition state?

We have added a definition for preleptotene, to clarify that this stage represents G1 and S phases prior to meiotic prophase (line 113-114).

5- In figure 3D the authors state that the small difference in RNA expression is consistent with a role in RNA destabilization. I'm curious why the authors conclude that the mechanism is at the RNA level and not just at the protein level? There are many reasons why a protein expression could be altered. Why do the authors choose this mechanism? Because the mechanism is completely speculative, the authors could expand on this.

We have added clarification that MEIOC was found to destabilize *Ccna2* during spermatogenesis, but other mechanisms may also contribute to the change in CCNA2 protein levels in oogenic cells (lines 232-235).

6- I found the use of full sentences in parenthesis distracting.

We have removed the parentheses (lines 54-57).

7- It seems like this mechanism is similar to how male meiotic entry is also regulated. This should be discussed in the discussion (or if anything is different).

We have included text to highlight similarities and differences between meiotic entry during oogenesis and spermatogenesis (lines 485-510).

8- In studies where the authors counted edu or other labeling, a total number of animals and cells

should be included in the legends.

Number of cells and animals have been added to figure panels or legends (Figure 1C, 3B, 3D).

9- In Edu incorporation, the authors indicate collecting embryos. At these stages, aren't they a fetus? 10- Line 534- I don't usually see the word "cover slipped" as a verb- lab jargon?

In the field of mouse development, "embryo" is commonly used to refer to all in utero stages. Consistent with this practice, we have kept the use of "embryo" in the text.

We have removed the use of "cover slipped" (lines 573-575).

Reviewer 2

The authors present data and re-analysis of genomic datasets to make conclusions regarding the mechanism by which mouse oogonia change from the mitotic DNA replication program to premeiotic DNA replication, and subsequently enter the meiotic gene expression program. While there is substantial data from the male side (spermatogonia>spermatocyte), oogenesis is harder to access and obviously has much different biology, so the question is relevant, important, and of substantial interest.

The experimental parts of this paper are essentially EdU labeling experiments of germ cells from embryonic ovaries of *Stra8* and *Meioc* knockouts, enabling an assessment of DNA replication activities. The bulk are from re-analysis of single cell and bulk RNA-seq datasets that were generated elsewhere, with a focus on transcript levels related to the cell cycle. Combining the bioinformatics and EdU experiments, the authors come up with a model of the regulatory circuitry summarized in Fig. 6 and summarized by the paper's title regarding the key role of MEIOC.

Overall, the topic is important and the conclusions drawn seem reasonable, albeit many aspects are correlative, not definitive. Gaining definitive molecular proof of the relationships in Fig. 6 would require extensive targeted experimentation, so in the meantime, I feel that this paper contributes a valuable conceptual framework.

SUGGESTIONS TO AUTHORS MAJOR POINTS

- with exception of assaying protein levels of *CCNA2*, no assessments of protein levels of cyclins were done nor were CDKs phosphorylation states assessed. Thus, nearly all the conclusions of cell cycle stages at different developmental time points were based on scRNA-seq data. Not sure how easy it would be to do Westerns on limiting quantities of sample.

We agree that measurements of additional Cyclin protein and CDK phosphorylation levels would be valuable data, but it is technically challenging to validate protein levels and phosphorylation states by Western blotting because of limited sample quantities in vivo.

We note that the scRNA-seq cell cycle analysis (Stuart et al., 2019) originates from a microarray analysis of bulk populations of synchronized cells identifying transcripts that are enriched at different phases of the cell cycle (Whitfield et al., 2002). This dataset was later used to identify cell cycle phase at the single cell level (Macosko et al., 2015) and then implemented within the Seurat package (Stuart et al., 2019).

To confirm cell cycle designations independent of the gene lists used by the Seurat package, we have also examined expression of *Mki67*, which is lowly expressed during G1 phase, and replication-dependent histones, whose transcripts are generated during late G1 and S phase (Figure S2E-F, lines 158-160).

- lines 273-276 "...we found that the STRA8-dependent genes (adjusted $P < 0.05$) exhibited significantly correlated effects in the MEIOC and STRA8 datasets (Figure 4C; Table S5). This confirms that the loss of MEIOC disrupts STRA8-MEIOSIN transcriptional activity." Correlations are not confirmation. Also, transcript abundance is not tantamount to transcriptional activity; one would have to measure transcription directly via methods such as run-on sequencing. In the absence of such data (I don't know if it exists), there should be ample caveats stated. Also, the section on the role of BMP signaling on MEIOC is also correlative, not regulation questions.

We have clarified this in the text, and have replaced “STRA8-MEIOSIN transcriptional activity” with “STRA8-MEIOSIN activity”. We agree that correlation alone does not provide molecular insights. Prior work has demonstrated that STRA8 and MEIOSIN are both required for the meiotic G1/S phase transition. Earlier in the manuscript, we demonstrate the *Meioc* KO exhibits prolonged mitotic cycling before meiotic G1/S phase transition and reduced *Meiosin* expression (Figure 1C-D, 4A). Given this context, correlation between the MEIOC and STRA8 datasets supports the model that the loss of MEIOC results in reduced STRA8-MEIOSIN activity (lines 295-297).

Based on our data, it is unclear if reduced *Meiosin* expression in the *Meioc* knockout is due to the delayed meiotic G1/S phase transition or more directly tied to the loss of MEIOC activity. We have added text to clarify this point (lines 485-496).

We agree that in vitro gametogenesis will be a valuable system to further develop this model, and we are working to establish this model in our lab.

- unlike studies of male meiosis, the authors do not perform analyses of direct binding by STRA8 to genes they conclude that are directly regulated by STRA8/MEIOSIN, or (for example), look at ATAC-seq data in WT vs mutants.

We have added a new bioinformatic analysis. Using STRA8 ChIP-seq data from the testes enriched for preleptotene spermatocytes (Kojima et al., 2019), as well as our list of STRA8-upregulated genes in the fetal ovary (Figure 4B), we have identified genes that are activated by STRA8 in oocytes. We then demonstrated that these STRA8-activated genes are enriched among MEIOC-upregulated genes (Figure 4C, lines 279-289).

MINOR POINTS

- Lines 88-90. For the non-expert reader, should specify that "oogenic cells" are defined as those that are DDX4/MVH-positive. It wouldn't hurt to say a few words about this protein's temporal expression pattern. Similarly for Pou5f1 (relevant for the section beginning on line 138).

We have added text to clarify use of DDX4 immunostaining (lines 89-91) and the Pou5f1 reporter (lines 151-152).

- Lines 98-101. It isn't clear to me how the conclusion is reached that some oogenic cells "initiate some DNA synthesis but fail to complete meiotic DNA replication." Perhaps explain more clearly (rather than just pointing to papers) that pre-meiotic DNA replication can stop in early S-phase once started (in the absence of checkpoint activation). Because the data in Fig. 1 is scoring cells as being positive or negative, is it possible that weaker staining reflects EdU from DNA repair? Similarly, on lines 108-109, it isn't clear how cells would be doing any DNA replication at the G1/S phase transition. Are the authors hypothesizing that such cells activate some licensed origins but never reach the restriction point?

We have clarified our rationale in the text and conclude that “in the absence of *Stra8*, the majority of oogenic cells arrest at the meiotic G1/S phase transition but some cells slip remains unclear whether the early S phase *Stra8* knockout oogenic cells are the result of an activated G1 checkpoint leaking into S phase or an inactive G1 checkpoint with activation of the S phase checkpoint (lines 450-458).

We have added Figure S1 to show that EdU is not incorporated in cells undergoing DNA repair during meiotic prophase I (lines 84-88). Thus, the EdU-positive population we have identified represents oogenic cells in S phase.

- line 277 "As MEIOC exhibits reduced, but not complete loss of, Meiosin gene expression..." Do the authors mean MEIOC absence?

We have updated the text accordingly (line 298).

- Fig. 6 Suggest that in the *Stra8* KO condition, put a slash through "cell cycle gene expression" since E2F will be inhibited by Rb1 in this scenario, if I understand correctly.

This is correct - in the *Stra8* KO, RB1 continues to repress E2F and consequently prevents E2F-mediated activation of cell cycle gene expression. To enhance the readability of the molecular networks in Figure 6, we have used grey to designate

regulation that is inactive due to upstream control.

Reviewer 3

Proper initiation of meiosis requires both the programmatic switch from mitotic to meiotic cell division and the activation of meiosis-specific genes. In previous studies, the authors demonstrated that MEIOC plays a crucial role in post-transcriptional regulation during male meiotic initiation. In this study, they further show that MEIOC is also required for the initiation of the mitosis-to-meiosis transition in oocytes. In addition to previously reported phenomena observed in males, the authors uncover female-specific features, such as the involvement of BMP signaling in regulating *Meioc* expression. These findings provide valuable insights and contribute significantly to the field. The data presented are generally consistent with the genetic network model proposed by the authors.

However, the main concern with this manuscript lies in the limited ability to infer molecular mechanisms based on the bulk RNA-sequencing data, which were obtained from a single time point. Incorporating data that offer mechanistic insights—such as time-course analyses, single-cell transcriptomics, or functional assays—would substantially enhance the impact of this study.

SUGGESTIONS TO AUTHORS

Major points to address:

Lines 248-314:

The authors discuss the effect of MEIOC on STRA8-MEIOSIN function based on bulk RNA-seq data collected at a single developmental time point. However, as the authors themselves acknowledge, *Meioc*-KO oocytes exhibit delayed meiotic initiation, implying a shift in developmental timing. Thus, it remains unclear whether the observed changes in Meiosin expression reflect a direct regulatory effect by MEIOC, or are instead a consequence of this developmental delay. The current data do not allow the authors to definitively conclude that MEIOC directly regulates STRA8-MEIOSIN activity. This issue should be carefully reconsidered and appropriately qualified in the manuscript.

We agree with the reviewer on this point - we do not know if delayed upregulation of *Meiosin* in the *Meioc* knockout is due to delayed entry into meiosis or more directly tied to the loss of MEIOC activity. We have added text to clarify this point (lines 485-496).

Lines 316-392:

The authors discuss the regulation of *Meioc* by BMP signaling. While there is little doubt that BMP signaling influences *Meioc* expression, it remains unresolved whether this regulation is direct or mediated indirectly via BMP-induced changes in cellular context. Clarifying this point—or at least explicitly discussing the possibilities—would strengthen the interpretation and is especially relevant when considering sex-specific differences in the regulation of *Meioc* and meiotic initiation.

We have added text stating that is not known whether BMP signaling directly or indirectly upregulates *Meioc* gene expression (lines 464-469).

Lines 933-936:

The labeling "Meiosis-associated genes upregulated by MEIOC and STRA8" in Fig4D is not consistent with "Fraction of meiosis-associated genes upregulated by MEIOC and BMPR1A" in the legend.

We have updated the figure legend for Figure 4D.

Second decision letter

MS ID#: dev.205037R1

MS TITLE: MEIOC prevents continued mitotic cycling and promotes meiotic entry during mouse oogenesis

AUTHORS: Esther G. Ushuhuda, Jenniluyun T. Nguyen, Natalie G. Pfaltzgraff, Shelbie M. Wenner, Matthew Kofron and Maria M. Mikedis

Dear Dr Mikedis,

I am happy to tell you that your manuscript has been accepted for publication in Development, pending our standard publication integrity checks.

Reviewer 1

Advance summary and potential significance to field

I remain excited about the information provided by the authors to this field. I find that they addressed my concerns appropriately.

Reviewer 2

Advance summary and potential significance to field

The reviewers have responded well to the reviews. I have no remaining concerns.

Reviewer 3

Advance summary and potential significance to field

The revisions adequately address all of my concerns.

This paper is outstanding in its discovery that MEIOC prevents continued mitotic cycling prior to meiotic entry, and that MEIOC regulates the STRA8-MEIOSIN axis and is involved in meiotic initiation in oogonia, thereby establishing important links to previous studies on STRA8 and MEIOSIN (Kojima et al 2019, Ishiguro et al 2020, Shimada et al 2023). Thus, this work makes a major contribution to our understanding of meiotic initiation in oogenic cells.

Comments for the author

However, I do not think the evidence for BMP signaling is strong enough to justify the authors' statement. Therefore, it would be better to remove the sentence from abstract, "We also demonstrate that BMP signaling halts mitotic cycling and promotes meiotic entry by upregulating MEIOC" (lines 11-12).

Kei-ichiro Ishiguro